# CHECKMATE! WATERMARKING GRAPH DIFFUSION MODELS IN POLYNOMIAL TIME

**Roberto Gheda**
TU Delft
r.gheda@tudelft.nl

**Abele Mălan**
University of Neuchâtel
abele.malan@unine.ch

**Robert Birke**
University of Turin
robert.birke@unito.it

**Maksim Kitsak**
TU Delft
m.a.kitsak@tudelft.nl

**Lydia Chen**
University of Neuchâtel
yiyu.chen@unine.ch

## ABSTRACT

Watermarking provides an effective means for data governance. However, conventional post-editing graph watermarking approaches degrade the graph quality and involve NP-hard subroutines. Alternatively, recent approaches advocate for embedding watermarking patterns in the noisy latent during data generation from diffusion models, but remain uncharted for graph models due to the hardness of inverting the graph diffusion process. In this work, we propose `CheckWate`: the first watermarking framework for graph diffusion models embedding checkerboard watermark and providing polynomial time verification. To address NP-completeness due to graph isomorphism, `CheckWate` embeds the watermark into the latent eigenvalues, which are isomorphism-invariant. To detect the watermark through reversing the graph diffusion process, `CheckWate` leverages the graph eigenvectors to approximately dequantize the discrete graph back to the continuous latent, with theoretical guarantees on the detectability and dequantization error. We further introduce a latent sparsification mechanism to enhance the robustness of `CheckWate` against graph modifications. We evaluate `CheckWate` on four datasets and four graph modification attacks, against three generation time watermark schemes. `CheckWate` achieves remarkable generation quality while being detectable under strong attacks such as isomorphism, whereas the baselines are unable to detect the watermark. Code available at: https://github.com/r-gheda/checkwate.

## 1 INTRODUCTION

Watermarking is a long-established method for data owners to verify the ownership of various data types (Cox et al., 2002) and it has recently been adapted to verify synthetic data from generative models (Yang et al., 2024). While graphs are used extensively for modeling real-world applications (Simonovsky & Komodakis, 2018) and synthetic graphs are increasingly adopted for knowledge discovery (Jo et al., 2023), a significant gap exists in watermarking solutions for them, especially for synthetic graphs. The conventional approaches (Zhao et al., 2015; Eppstein et al., 2016) embed watermarks in graphs via post-editing, which reduces the graph quality and requires exponential time verification. In contrast, modern watermarking methods (Yang et al., 2024; Wen et al., 2023; Zhu et al., 2025) embed the watermark at sampling-time in the noisy latents of diffusion models. These methods have the advantages of quality conservation and robust detectability at the expense of inverting the diffusion process. However, their effectiveness has only been validated on modalities other than synthetic graphs.

In contrast to images and tables, graphs can be represented in multiple adjacency matrices via isomorphism, making differentiating (un)watermarked graphs hard. Fig. 1 illustrates an example of how node indices can be arbitrarily swapped without changing the structure of the graph. The **Graph Isomorphism** (GI) problem, i.e., determining whether two graphs are isomorphic, is one of the few unresolved questions in complexity theory, as it is not known whether it can be solved

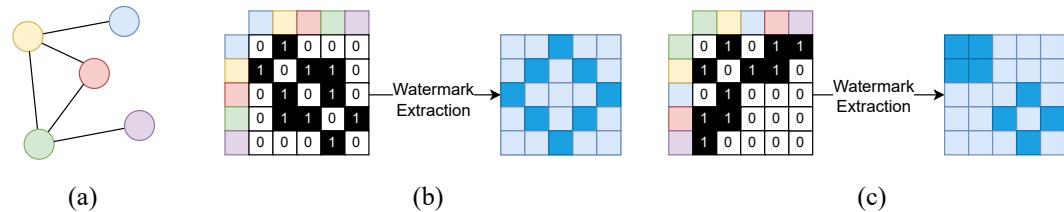

Figure 1: An example of how Graph Isomorphism disrupts watermark patterns. (a) A graph $G$ (b) A valid representation of $G$, the watermarking pattern can be successfully extracted. (c) Using another valid representation (*isomorphism*) of $G$ disrupts the watermarking pattern.

in P (Babai, 2016; Grohe & Schweitzer, 2020). While quasi-polynomial algorithms exist for small instances (Grohe & Neuen, 2021), GI remains computationally infeasible for large graphs. Furthermore, when graphs are modified through addition or deletion of edges, solving GI requires addressing the *Graph Edit Distance* (GED) problem (Bunke, 1997), which is well-known to be NP-hard.

The discrete nature of graphs presents another unique challenge to watermarking, as it complicates watermark verification, particularly when coupled with the GI problem. As graphs are often represented via binary adjacency matrices, graph diffusion models, such as GruM (Jo et al., 2023), require moving from the continuous space to the discrete one through a quantization step. When watermarks are embedded in latents, the verification of watermarks needs to invert any graph back to its latents, by first **dequantizing** the graph. Inverting this step requires matching the generated graph with its corresponding dequantized version. This is computationally infeasible due to the complexity of the GI and GED problems.

We introduce `CheckWate`, the first framework for watermarking graph diffusion models with a robust verification in polynomial time. `CheckWate` consists of three key components. (i) A **checkerboard watermark** technique that enables embedding watermarking on the noisy latent eigenvalues at sampling-time. Since eigenvalues are isomorphism-invariant, this allows us to embed and extract the watermark in polynomial time with no loss of generalization and without relying on any approximation. (ii) An **approximate dequantization** mechanism that enables the transition from the discrete domain of the data to the continuous space of diffusion, thus accurate latent reconstruction and watermark verification. (iii) A **robust detection mechanism** that further improves watermark detection robustness, especially for mitigating false positive verification. Drawing upon matrix sparsification theory, we identify reconstruction errors within the noisy latent and impose constraints on the distribution of their eigenvalues. Our work brings the following contributions:

- `CheckWate` is a non-blind graph watermark algorithm with verification in polynomial time, circumventing NP-hardness from graph isomorphism.
- `CheckWate` detects the presence of a watermark by accurately inverting graph diffusion, via an approximate dequantization mechanism with a theoretical error bound.
- `CheckWate` robustness against post-editing attacks is enhanced by a latent sparsification mechanism.
- Extensive evaluation of `CheckWate` on the graph quality and watermark detectability on four datasets and four graph attacks.

## 2  RELATED WORK

**Graph Diffusion Models** Graph synthesizers have been of high interest for the scientific community in the past years. Graph diffusion models generate data starting from random (symmetric) noise. Then, a trained neural network iteratively predicts the probability distribution of clean graphs and moves toward such distribution via steps of *Denoising Diffusion Probabilistic Model* (DDPM) (Ho et al., 2020). Depending on the model, this diffusion can either happen on the discrete or the continuous space. GruM (Jo et al., 2023) introduced a novel diffusion model based on *Denoising Diffusion Bridge Models* (DDBM) (Zhou et al., 2023) that performs diffusion on the latent continuous space and achieves state-of-the-art generative performance. The denoising process of GruM is proven to converge to the discrete space of the graphs adjacency matrix up to quantization.

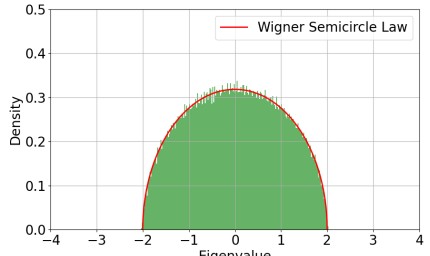 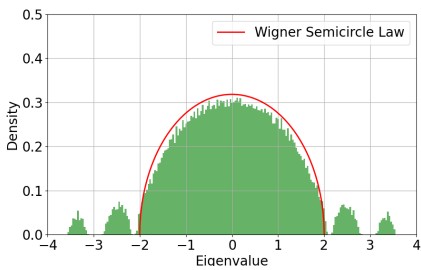

(a) Eigenvalues of Gaussian Orthogonal Ensembles.    (b) Eigenvalues of Checkerboard Ensembles.

Figure 2: Distribution of eigenvalues of random matrices. (a) The eigenvalues of Gaussian Orthogonal Ensembles ($N = 500$) follow the Wigner semicircle law (*bulk*). (b) Checkerboard Ensembles ($N = 500, k = 450, W = (\pm 1.5, \pm 2.5)$) have $N-k$ eigenvalues outside the semicircle (*blip*).

**Watermarking Synthetic Data** Watermarking is one of the key techniques used for verifying the ownership of synthetic data. Effective watermarking requires imperceptibility and robust detectability. The watermark signals can be embedded during the model training, sampling-time, or even post-data generation (He et al., 2024), having different degrees of tradeoff between the data quality and robustness. Post-editing watermarks are the conventional methods applied on real graphs, leading to significant quality degradation. To avoid this problem, newer methods such as TreeRing (Wen et al., 2023) and Gaussian Shading (Yang et al., 2024) embed a pattern into the latent space. However, TreeRing directly disrupts the Gaussian distribution of noise, limiting the randomness of sampling and resulting in affecting model generative performance. Moreover, these techniques require to undergo an implicit diffusion model such as *Denoising Diffusion Implicit Model* (DDIM) (Song et al., 2020) or *Denoising Bridge Implicit Model* (DBIM) (Zheng et al., 2024) in order to be accurately inverted. Which are the implicit versions of DDPM and DDBM respectively (details in Appendix A).

Techniques similar to Gaussian Shading have been used to extend applicability to domains different from images such as tabular data (Zhu et al., 2025) and time series (Soi et al., 2025). Nevertheless, they still require to verify a pattern within the latent. This prevents the application of these methods to the domain of graphs, as graph isomorphism enables representing the data in $N!$ ways.

**Graph Watermarking** The prior art on watermarking graphs centers on real graph, thus being post-editing approaches. The long standing challenge is to determine two graphs are isomorphic and only quasi-polynomial time solutions exist. Eppstein et al. (2016) further shows that when undergoing adversarial attack, solving isomorphism requires to address the more complex graph edit distance problem, which is NP-hard. Specifically, Zhao et al. (2015) and Eppstein et al. (2016) provide post-editing applications for non-blind graph watermarking. However, Zhao et al. (2015) makes assumptions on the graph node degree distribution to make the GED problem tractable, while remaining exponential in the cost and not being applicable to all graphs. Eppstein et al. (2016) provides an approximate NP-complete solution to GI in exponential time. Recently, KGMark (Peng et al., 2025) addresses post-editing watermarking for knowledge graphs only, whereas Bourrée et al. (2025) proposes watermark graphs in the Fourier spectrum assuming nodes are uniquely labeled. All of these works make strong assumptions on the data or the attacks that can be performed by the adversaries.

## 3 CHECKWATE

We start with preliminaries on random matrix theory in Section 3.1, before introducing the Check-Wate methodology shown in Fig. 3. In Section 3.2, we first delve into the watermark injection and detection mechanisms (steps ①, ⑥). Then, in Section 3.3, we discuss our proposed method for inverting quantization (steps ③, ④). Finally, in Section 3.4, we cover our error mitigation strategy, which prevents false positives under heavy adversarial perturbations. Further, we provide Check-Wate pseudocode in Appendix D.

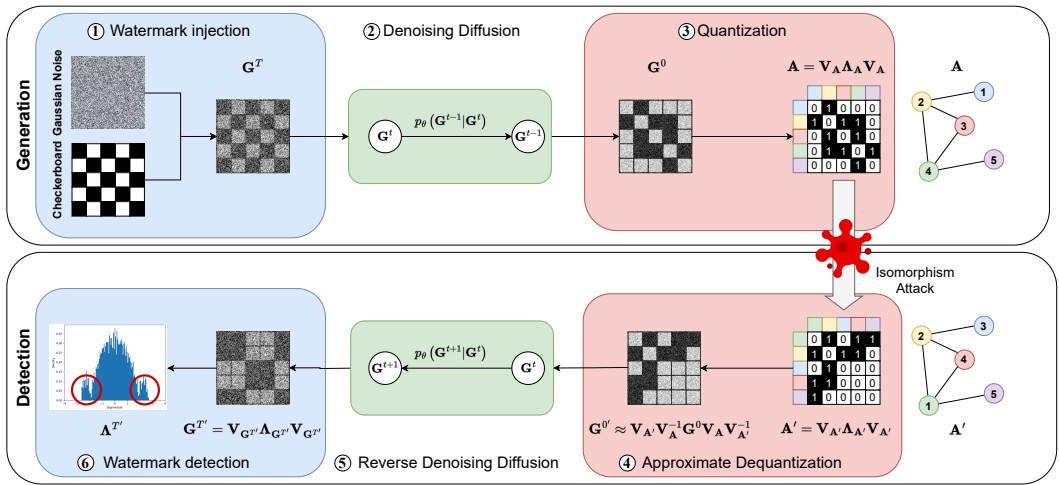

Figure 3: Pipeline of `CheckWate`. ① **Watermark injection**. ② **Denoising diffusion**. ③ **Quantization**. ④ **Dequantization**. ⑤ **Reverse denoising diffusion**. ⑥ **Watermark detection**.

## 3.1 PRELIMINARIES ON RANDOM MATRIX THEORY

Graph diffusion models such as GruM rely on noisy latent variables modeled as Gaussian Orthogonal Ensembles (GOEs) (Anderson et al., 2010), i.e., symmetric Gaussian random matrices. The following explains the fundamental properties of random matrix theory used in our framework.

**Spectral measure** The spectrum of graphs and random matrices are essential in capturing information on the graph structure (Van Mieghem, 2023). Eigenvalue distributions are typically divided into two regimes: the *bulk* (of order $\mathcal{O}(\sqrt{N})$) and the *blip* (of order $\mathcal{O}(N)$). For GOEs of size $N \times N$, all eigenvalues lie within the bulk. More precisely, when considering the normalized ensemble $\mathbf{X}_N = \mathbf{X}/\sqrt{N}$, the limiting spectral distribution $\rho$ follows the *Wigner semicircle law* (Anderson et al., 2010) with radius $R = 2$. Fig. 2a shows the distribution of eigenvalues from GOEs.

**Checkerboard Ensembles** $(k, W)$-checkerboard ensembles were introduced by Chen et al. (2020) as a generalization of Burkhardt et al. (2018). Let $\mathcal{N}(0, 1)$ be the standard normal distribution.

**Definition 3.1.** *Fix $k \in \mathbb{N}$ and a $k$-tuple of real numbers $W = (W_1, \ldots, W_k)$, then the $N \times N$ $(k, W)$-checkerboard ensemble ($(k, W)$-CBE) is the ensemble of matrices $\mathbf{C} = (\mathbf{C}_{ij})$ given by:*

$$\mathbf{C}_{ij} = \mathbf{C}_{ji} = \begin{cases} \mathcal{N}(0,1) & i \not\equiv j \mod k \\ W_u & i \equiv j \equiv u \mod k, \text{ with } u \in \mathbb{Z}_k \end{cases}$$

Checkerboard ensembles have $k$ eigenvalues in the bulk, while the remaining $N - k$ eigenvalues are in the blip. More precisely, for each $W_i \in W$, if $W_i$ appears $k_i$ times in the ensemble, $k_i$ eigenvalues are of magnitude $NW_i/k + \mathcal{O}(\sqrt{N})$. Fig. 2b shows the eigenvalue distribution from a Checkerboard Ensemble. For high enough $k$, Checkerboard ensembles allow to apply significant changes to the ensemble spectrum while forcing limited modifications from regular Gaussian noise. This allows strong detectability while preserving generation quality.

## 3.2 CHECKERBOARD WATERMARK

The key hurdle to embed and detect the watermark pattern on graphs lies in the inherent ambiguity of graph representation, illustrated in Fig. 1. In contrast, taking the image as an example, pixel positions cannot be freely permuted without severely degrading visual quality—or even altering the picture's entire meaning. This enables the application of pattern-based watermarks onto their latents, such as Gaussian Shading (Yang et al., 2024) and TreeRing (Wen et al., 2023). Similarly, tabular data can leverage column unambiguity to apply row-level watermark detection (Zhu et al., 2025).

To address this, `CheckWate` explores the properties of graph eigenvalues in both embedding and detection phases. To embed the watermark, step ①, we insert the checkerboard pattern in the noisy latent of synthetic graph at sampling-time. This enables the presence of eigenvalues in the blip, which are not expected in regular GOE, while applying minimal changes to the latent. Specifically, applying minimal changes to the GOE, ensures that the generation quality is preserved. To detect the checkerboard pattern, we first revert the graph to its noisy latent and then inspect the presence of eigenvalues in the blip regime. The advantages of leveraging the eigenvalues of the latent are two-fold: polynomial time computation and isomorphism invariance. This allows to bypass the GI problem and detect watermark in $\mathcal{O}(N^3)$ with no approximation nor generalization loss.

**Watermark injection** ① We start graph generation via a random symmetric matrix $\mathbf{G}^T$ in the latent space. We define $\mathbf{G}^T$ as checkerboard ensemble with $W = (W_1, \ldots, W_k)$ an array of size $k$:

$$\mathbf{G}_{ij}^T = \mathbf{G}_{ji}^T = \begin{cases} \mathcal{N}(0,1) & i \not\equiv j \mod k \\ W_u & i \equiv j \equiv u \mod k, \text{ with } u \in \mathbb{Z}_k \end{cases} \tag{1}$$

**Parameter tuning** Both $k$ and $W$ are hyperparameters that determine the magnitude and frequency of eigenvalues in the blip. They allow to balance generation quality with watermark detectability. In general, checkerboard ensembles have $N-k$ eigenvalues in the blip. Thus, a lower value of $k$ increases the watermark strength. Each of the eigenvalues in the blip has magnitude $NW_u/k + \mathcal{O}(\sqrt{N})$. Thus, a larger $W_u/k$ increases the watermark detectability. At the same time, having a lower $k$ increases the amount of non-Gaussian entries and a larger $W_u/k$ increases the deviation from Gaussian noise. Both affect negatively generation quality creating a tradeoff with detectability: the larger $W_u$ and the smaller $k$ the stronger the watermark and the lower the quality.

**Watermark detection** ⑥ To detect the watermark, we reverse the diffusion process and reconstruct a noisy latent $\mathbf{G}^{T'} \approx \mathbf{G}^T$. We detail the reconstruction process in Section 3.3. Then, we compute the eigenvalues of the normalized $\mathbf{G}_N^{T'} = \mathbf{G}^{T'}/\sqrt{N}$. If $\mathbf{G}^{T'}$ is reconstructed from a watermarked graph, we expect $N-k$ eigenvalues to fall in the blip. Thus, we measure the (absolute) largest $N-k$ eigenvalues and expect them all to be $\gg 2$. If $\mathbf{G}^{T'}$ is not reconstructed from a watermarked graph, $\mathbf{G}^{T'}$ is a GOE and we expect all of its eigenvalues to fall in the bulk, i.e., the largest $N-k$ eigenvalues are $\leq 2$. We derive the expected difference between the computed score of a watermarked graph $\mathbf{G}^W$ and a non-watermarked one $\mathbf{G}^{NW}$ to measure `CheckWate` detectability:

**Theorem 3.1** (Watermark Detectability). *Let $\mathbf{G}_N^{NW}$ be a normalized Gaussian Orthogonal Ensembles. Let $\mathbf{G}_N^W$ be a $N$ normalized $(k, W_u)$-Checkerboard Ensembles. Let $\lambda_i(\mathbf{G})$ the $i$-th largest eigenvalue of $\mathbf{G}$. Detectability of* `CheckWate` *is defined as:*

$$\mathbb{E}_{\mathbf{G}_N^W \sim (k, W_u)\text{-CBE}} \left[ \frac{\sum_{i=1}^{N-k} \lambda_i\left(\mathbf{G}_N^W\right)}{(N-k)} \right] - \mathbb{E}_{\mathbf{G}_N^{NW} \sim GOE} \left[ \frac{\sum_{i=1}^{N-k} \lambda_i\left(\mathbf{G}_N^{NW}\right)}{(N-k)} \right] = \frac{\sqrt{N}W_u}{k} + \mathcal{O}(1) - \mathcal{O}(k^2) \tag{2}$$

We prove Theorem 3.1 in full in Appendix C.1. Equation 2 reinforces that the strength of `Check-Wate` watermark is proportional to $W_u$ and inversely proportional to $k$.

## 3.3 Approximate Dequantization

Unlike other modalities, graphs are commonly represented via binary adjacency matrices. Hence, generating graphs requires transitioning from the continuous space of denoising diffusion models to the discrete space of binary adjacency matrices through quantization, step ③. Inverting this step amounts to matching an edited graph to its dequantized matrix, which reduces to the NP-complete GI and GED problems. We further stress that it is not possible to accurately reconstruct the noisy latent without taking into account quantization and demonstrate this in Table 1 of Section 4.

To overcome this, we provide a dequantization method, step ④, that leverages fundamental properties of eigenvectors under permutation to approximately reconstruct the dequantized graph latent in $\mathcal{O}(N^3)$. We discuss next the basic properties of this approximation and derive the exact reconstruction error under the Frobenius norm.

**Graph quantization** ③ After diffusion, the obtained graph $\mathbf{G}^0$ is composed of continuous values. Thus, we require to quantize $\mathbf{G}^0$ to a 0-1 graph adjacency matrix. After this step, we obtain the

quantized adjacency matrix $\mathbf{A}$:

$$\mathbf{A} = \text{quantize}(\mathbf{G}^0), \mathbf{A}_{ij} \in \{0, 1\}$$

**Approximate dequantization** ④ For a given graph $\mathbf{A}'$, we need to verify the existence of a checkerboard watermark. First, a graph $\mathbf{A}'$ can be a different, i.e., permuted, representation of $\mathbf{A}$:

$$\mathbf{A}' = \mathbf{P}\mathbf{A}\mathbf{P}^{-1}$$

where $\mathbf{P}$ is an unknown permutation matrix. To accurately reconstruct the original noise $\mathbf{G}^T$ in continuous space, we need the dequantized version of $\mathbf{A}'$, i.e., $\mathbf{G}^{0'} = \mathbf{P}\mathbf{G}^0\mathbf{P}^{-1}$. Eigenvectors of permuted graphs permute accordingly up to change of eigenbasis, i.e., $\mathbf{V}_{\mathbf{A}'} = \mathbf{P}\mathbf{V}_{\mathbf{A}}\mathbf{Q} \approx \mathbf{P}\mathbf{V}_{\mathbf{A}}$ where $\mathbf{V}_A$ are the eigenvectors of $\mathbf{A}$, and $\mathbf{Q}$ is a block-diagonal matrix that maps the eigenbasis of $\mathbf{A}$ to the one of $\mathbf{A}'$. More precisely, $\mathbf{Q} = \text{diag}(\mathbf{Q}_1, \ldots, \mathbf{Q}_m)$ with $m$ the number of distinct eigenvalues of $\mathbf{A}$ and each $\mathbf{Q}_i$ being a rotation square matrix as large as the algebraic multiplicity of eigenvalue $\lambda_i$, $e(\lambda_i)$. We leverage this property and combine some algebraic simplification to obtain the approximation of the permuted dequantized graph $\mathbf{G}^{0'}$:

**Theorem 3.2** (Approximate Dequantization). *Let $\mathbf{A}'$ be a permutation of $\mathbf{A}$ based on a permutation matrix $\mathbf{P}$. Let $\mathbf{G}^{0'}$ be a permutation of $\mathbf{G}^0$ on the same permutation matrix $\mathbf{P}$. Then, $\mathbf{G}^{0'}$ can be accurately approximated as:*

$$\mathbf{G}^{0'} \approx \mathbf{V}_{\mathbf{A}'}\mathbf{V}_{\mathbf{A}}^{-1}\mathbf{G}^0\mathbf{V}_{\mathbf{A}}\mathbf{V}_{\mathbf{A}'}^{-1} \tag{3}$$

We prove Theorem 3.2 in Appendix C.2. In the special case in which all the eigenvalues are distinct, $\mathbf{Q}$ is the identity matrix and equality for Equation 3 always holds. Further, the error given by the approximation in Equation 3 can be explicitly derived as a function of $\mathbf{Q}$:

**Theorem 3.3** (Reconstruction Error). *Let $\mathbf{G}^0$ be an orthogonal matrix and $\mathbf{G}^{0'}$ computed as in Equation 3. Then the reconstruction error can be derived as:*

$$\|\mathbf{G}^{0'} - \mathbf{G}^0\|_F = \sum_{r,s=1}^{m} \left(\|\mathbf{Q}_r\{\mathbf{V}_{\mathbf{A}}^{-1}\mathbf{G}^0\mathbf{V}_{\mathbf{A}}\}_{r,s}\mathbf{Q}_s^{-1} - \{\mathbf{V}_{\mathbf{A}}^{-1}\mathbf{G}^0\mathbf{V}_{\mathbf{A}}\}_{r,s}\|_F^2\right)^{1/2} \tag{4}$$

We prove Theorem 3.3 in Appendix C.3. In general, the error increases with eigenvalue multiplicity: the higher the multiplicity, the higher the error. For a graph $\mathbf{G}$, the number of distinct eigenvalues is $e(\mathbf{G})$ and satisfies $e(\mathbf{G})d + 1$. Therefore, maximum multiplicity is always bounded by $N - d$. Graphs sampled from the Barabasi-Albert model (Albert & Barabási, 2002) have $d \approx \frac{\ln(N)}{\ln(pN)}$.

**Diffusion inversion** From here on, the diffusion process can be easily inverted following one of the known paradigms of DDIM or DBIM (Wen et al., 2023; Yang et al., 2024; Zhu et al., 2025; Soi et al., 2025). We detail diffusion implicit models and their inversion in Appendix A.

**Authorship identification via hash and sign** To enable identification of the watermark author we use digital hash-based signature (Srivastava et al., 2023). Hash-based signatures are a wide-spread cryptographic building block that provides authenticity, unforgeability, and undeniability (Srivastava et al., 2023). The watermark author can hash-sign the key $\mathbf{K} = \mathbf{V}_{\mathbf{A}}^{-1}\mathbf{G}^0\mathbf{V}_{\mathbf{A}}$ which is then used to enable reversibility of the diffusion process and reconstruct the noisy latent to extract the watermark. We stress that accurately computing the dequantized matrix $\mathbf{G}^{0'}$ is essential to enable watermark detectability as we demonstrate in our experiments in Section 4.

### 3.4 Robust Detection via Latent Sparsification

Even under perfect reversal of the diffusion process, the reconstructed latent $\mathbf{G}^{T'}$ might be subject to several perturbations due to approximation errors stemming from Equation 3 and adversarial perturbations on the graph $\mathbf{A}$. Unlike watermark reconstructions in other modalities, where errors remain largely localized, eigenvalues encode global structural dependencies within the matrix and are thus far more sensitive to such disturbances. Consequently, the eigenvalues of the perturbed GOE might fall out of the bulk regime, i.e., $> 2$, and lead to false positive behavior. To prevent this, we apply a simple yet efficient robustness enhancing mechanism that replaces entries that were unlikely generated in the original noisy latent:

$$\mathbf{G}_{ij}^{T'} = \begin{cases} \mathbf{G}_{ij}^{T'} & \max\left(\phi(\mathbf{G}_{ij}^{T'}), \delta(\mathbf{G}_{ij}^{T'})\right) > \theta \\ 0 & \text{otherwise} \end{cases} \tag{5}$$

Where $\phi(\cdot)$ and $\delta(\cdot)$ are the probability density functions of a normal Gaussian and a Dirac distribution, centered at 0 and $W$ respectively. $\theta$ is a threshold parameter that determines the tolerance of the anomaly detection mechanism. This leads to replacement of entries with values unlikely belonging to the original noisy latent with zero entries, i.e., sparsification. Sparsifying the latent allows us to better control the behavior of the eigenvalues and prevent their explosion outside of the bulk.

**Rationale** After this process, $\mathbf{G}^{T'}$ is a sparse GOE. Let $q$ be the number of non-zero elements per matrix row. When $q = N$, $\mathbf{G}^{T'}$ is a GOE and its eigenvalues follow the Wigner semicircle. When $q < N$, most of the eigenvalues lie in the bulk, with a higher density around zero (Evangelou, 1992). More precisely, for small values of $q$, the density of eigenvalues $\rho$ is:

$$\rho(\lambda) \propto \frac{1}{|\lambda| \log\left(|\lambda|\right)^3} \tag{6}$$

as $\lambda \to 0$ (Evangelou, 1992). While exponential tails develop outside of the bulk domain, for high enough $q$ these eigenvalues are rare and close enough to the bulk not to create any practical problem for watermark detection even under heavy perturbations. We demonstrate this qualitatively in Appendix H and with numerical results in Table 2.

## 4 EVALUATION

We consider four **datasets** from prior work on graph diffusion models (Jo et al., 2023; Vignac et al., 2022; Martinkus et al., 2022): Planar, Tree, Stochastic Block Model (SBM), and Proteins (Dobson & Doig). Because no prior semantic watermarking methods exist for graph diffusion, we adapt two state-of-the-art **baselines** to the graph domain: *Gaussian Shading* (Yang et al., 2024) and *TreeRing* (Wen et al., 2023). We additionally design a graph-specific baseline, *Bipartite*, which is graph-invariant but severely compromises generative quality due to the high correlation of entries of the noisy latents. Implementation details for all baselines, including Bipartite, are in Appendix B. Finally, *None* serves as the non-watermarked reference.

To evaluate **generative performance**, we follow the setting of Jo et al. (2023). We measure the maximum mean discrepancy (MMD) of four graph statistics between the set of generated graphs and the test set: degree (Deg.), clustering coefficient (Clus.), count of orbits with 4 nodes (Orb.), and the eigenvalues of the graph Laplacian (Spec.). We also compute the percentage of valid, unique, and novel (V.U.N.) graphs for which the validity is defined as satisfying the specific property of each dataset. We evaluate **watermark detectability** via Z-score, which measures the distance between the mean score of watermarked and non-watermarked data normalized by the standard deviation.

### 4.1 GENERATIVE QUALITY AND WATERMARK DETECTABILITY

We run experiments with no attack on all four datasets. Results are showcased in Table 1. First, we can see that `CheckWate` achieves state-of-the-art **generative quality**. Namely, `CheckWate` is the best performing watermarking method 10 times out of 20, and second-best 9 times out of the remaining 10. The best baseline from the state-of-the-art is Gaussian Shading, which is a provably lossless watermarking. TreeRing and Bipartite, fall significantly behind, as they perform up to 10 times worse than the best watermarking method depending on the dataset and quality metric. Both `CheckWate` and Gaussian Shading achieve generative performance comparable to the one obtained without watermark. Under the Proteins dataset, `CheckWate` significantly outperforms other baselines including None. We suggest that the enhanced variance obtained via the checkered entries compensates the lack of randomness in the used implicit model.

For **detectability**, all methods achieve consistent results, except for TreeRing. Bipartite achieves the best Z-score, but lacks generative quality. Gaussian Shading and `CheckWate` have comparable Z-scores, except on Proteins where `CheckWate` significantly outperforms Gaussian Shading. Finally, we emphasize the detectability obtained with *No Dequantization*. Not applying a dequantization leads to an almost complete –if not complete– loss of the watermark even under no attack.

Table 1: Generative quality is reported as mean maximum discrepancy (MMD) and the ratio of valid, unique, and novel (V.U.N.) samples. Watermark detectability is evaluated via the Z-score, with results shown for both dequantized and non-dequantized graphs. **Bold** indicates the best result and underline the second best; arrows specify if lower or higher values are preferable. We note with a checkmark detectable Z-scores ($> 10$).

| Dataset | Watermark | Quality Metrics | | | | | Detectability (Z-Score) | |
|---|---|---|---|---|---|---|---|---|
| | | Deg. ↓ | Clus. ↓ | Orb. ↓ | Spec. ↓ | V.U.N. (%) ↑ | Dequant ↑ | No Dequant. ↑ |
| Planar $|V| = 64$ Synthetic | *None* | *0.0009* | *0.0373* | *0.0123* | *0.0078* | *72.5* | – | – |
| | Gaussian Shading | 0.0008 | **0.00367** | **0.0072** | 0.0093 | 62.5 | 57.6 ✓ | 5.9 |
| | TreeRing | 0.0104 | 0.1255 | 0.2122 | 0.0188 | 10 | 1.0 | 0.0 |
| | Bipartite | **0.0007** | 0.0412 | 0.0653 | 0.0101 | **67.5** | **4999** ✓ | 0.5 |
| | **CheckWate (ours)** | 0.0008 | 0.0418 | 0.0137 | **0.0078** | 67.5 | 67.6 ✓ | 0.7 |
| Tree $|V| = 64$ Synthetic | *None* | *0.0007* | *0* | *0.0001* | *0.0126* | *55* | – | – |
| | Gaussian Shading | **0** | **0** | 0.0001 | **0.0090** | **67.5** | 59.0 ✓ | 2.9 |
| | TreeRing | 0.0002 | **0** | 0.0002 | 0.0122 | 42.5 | 1.1 | 0.0 |
| | Bipartite | 0.0002 | **0** | 0.0002 | 0.0124 | 45 | **4968** ✓ | 0.0 |
| | **CheckWate (ours)** | 0.0004 | **0** | **0** | 0.0093 | 47.5 | 45.0 ✓ | 0.3 |
| SBM $44 \leq |V| \leq 187$ Synthetic | *None* | *0.005* | *0.0504* | *0.0439* | *0.0058* | *67.5* | – | – |
| | Gaussian Shading | 0.0105 | **0.0498** | 0.0629 | 0.0076 | **72.5** | 84.7 ✓ | 10.7 ✓ |
| | TreeRing | 0.0264 | 0.0612 | 0.1100 | 0.0112 | 52.5 | 0.6 | 0.0 |
| | Bipartite | 0.1178 | 0.6432 | 0.1984 | 0.0742 | 0 | **915.8** ✓ | 25.4 ✓ |
| | **CheckWate (ours)** | **0.0041** | 0.0520 | **0.0552** | **0.0050** | 52.5 | 86.8 ✓ | 0.0 |
| Proteins $100 \leq |V| \leq 500$ Real | *None* | *0.4315* | *0.5436* | *1.3283* | *0.2450* | – | – | – |
| | Gaussian Shading | 0.4358 | 0.5283 | 1.3332 | 0.2579 | – | 110.3 ✓ | 1.4 |
| | TreeRing | 0.4149 | 0.4137 | 1.3332 | 0.3009 | – | 1.5 | 0.0 |
| | Bipartite | 0.5114 | 1.0123 | 1.4257 | 0.5636 | – | **1724.0** ✓ | 2.2 |
| | **CheckWate (ours)** | **0.0473** | **0.2156** | **0.5986** | **0.0440** | – | 404.7 ✓ | 2.0 |

Table 2: Watermark detectability (Z-score) under perturbations. The higher the better. **Bold** denotes best, underlined denotes second-best. $\|e_{max}\|$ is average maximum graph eigenvalue multiplicity. We note with a checkmark detectable Z-scores ($> 10$).

| Dataset | Watermark | $\|e_{max}\|$ | Watermark Detectability (Z-Score ↑)) | | | | | | | | | |
|---|---|---|---|---|---|---|---|---|---|---|---|---|
| | | | Isomorphism | Edge Deletion | | | Edge Addition | | | Node Deletion | | |
| | | | | 5% | 10% | 20% | 5% | 10% | 20% | 5% | 10% | 20% |
| Planar | Gaussian Shading | (57.6 ✓) | 2.9 | 0 | 2.3 | 2.2 | 2.3 | 3.5 | 2.9 | 4.6 | 2.5 | 2.4 | 1.4 |
| | TreeRing | (1.0) | 2.8 | 0 | 0 | 0 | 0 | 0 | 0 | 0 | 0 | 0 | 0 |
| | Bipartite | (4999 ✓) | 3.0 | 548.5 ✓ | 499.0 ✓ | 477.2 ✓ | 605.4 ✓ | 211.6 ✓ | 116.8 ✓ | 39.8 ✓ | 448.5 ✓ | 528.6 ✓ | 541.7 ✓ |
| | **CheckWate (ours)** | (67.6 ✓) | 2.9 | 41.8 ✓ | 36.5 ✓ | 35.6 ✓ | 36.7 ✓ | 32.6 ✓ | 30.9 ✓ | 21.8 ✓ | 30.8 ✓ | 31.5 ✓ | 28.6 ✓ |
| Tree | Gaussian Shading | (59.0 ✓) | 8.6 | 0 | 2.8 | 2.3 | 1.2 | 3.1 | 1.9 | 1.7 | 2.3 | 1.7 | 1.1 |
| | TreeRing | (1.1) | 9.0 | 0 | 0 | 0 | 0 | 0 | 0 | 0 | 0 | 0 | 0 |
| | Bipartite | (4967.7 ✓) | 9.2 | 166.6 ✓ | 111.4 ✓ | 113.1 ✓ | 119.9 ✓ | 103.7 ✓ | 83.3 ✓ | 56.2 ✓ | 133.9 ✓ | 104.4 ✓ | 118.9 ✓ |
| | **CheckWate (ours)** | (45.0 ✓) | 9.3 | 31.3 ✓ | 28.8 ✓ | 31.1 ✓ | 30.5 ✓ | 34.9 ✓ | 35.7 ✓ | 31.8 ✓ | 24.3 ✓ | 33.3 ✓ | 23.8 ✓ |
| SBM | Gaussian Shading | (85.1 ✓) | 80.6 | 0 | 5.3 | 4.6 | 3.9 | 3.9 | 3.4 | 3.6 | 3.8 | 2.9 | 2.2 |
| | TreeRing | (0.5) | 80.8 | 0 | 0 | 0 | 0 | 0 | 0 | 0 | 0 | 0 | 0 |
| | Bipartite | (894.6 ✓) | 107.5 | 124.1 ✓ | 54.6 ✓ | 209.2 ✓ | 20.5 ✓ | 618.3 ✓ | 246.5 ✓ | 141.5 ✓ | 623.2 ✓ | 256.8 ✓ | 17.5 ✓ |
| | **CheckWate (ours)** | (86.8 ✓) | 80.7 | 60.8 ✓ | 50.4 ✓ | 19.5 ✓ | 15.6 ✓ | 18.8 ✓ | 10.6 ✓ | 11.1 ✓ | 32.8 ✓ | 24.4 ✓ | 9.7 |
| Proteins | Gaussian Shading | (119.3 ✓) | 305.4 | 0.1 | 0.7 | 0.6 | 0 | 0.8 | 0.7 | 0.3 | 0.7 | 0.3 | 0 |
| | TreeRing | (1.0) | 339.0 | 0 | 0 | 0 | 0 | 0 | 0 | 0 | 0 | 0 | 0 |
| | Bipartite | (1724.0 ✓) | 481.1 | 1636.7 ✓ | 1636.9 ✓ | 1636.9 ✓ | 1634.5 ✓ | 1636.8 ✓ | 1636.8 ✓ | 1634.3 ✓ | 1636.8 ✓ | 1636.8 ✓ | 1634.4 ✓ |
| | **CheckWate (ours)** | (404.7 ✓) | 267.8 | 128.4 ✓ | 174.7 ✓ | 112.0 ✓ | 92.2 ✓ | 166.3 ✓ | 117.7 ✓ | 59.7 ✓ | 152.4 ✓ | 71.1 ✓ | 46.2 ✓ |

## 4.2 Robustness to Graph Perturbations

We test watermark robustness under four graph-specific perturbations (Table 2): Isomorphism, Edge Addition, Edge Deletion, and Node Deletion, each applied at strengths from 5% to 20%. Check-Wate remains detectable under all experimented attacks, with the lowest Z-score (9.7) observed for SBM under 20% node deletion. Bipartite is consistently the strongest watermarking method but sacrifices generative quality. The state-of-the-art methods fail at achieving a statistically significant watermark most of the times as TreeRing never reaches a positive Z-score, and Gaussian Shading never surpasses a Z-score of $5.3$. Notably, under isomorphism, they both consistently achieve a Z-score of $0$, confirming that they are not graph invariant. We also report the average maximum eigenvalue multiplicity $\|e_{max}\|$, showing that generated graphs rarely have low multiplicity. This reinforces the strength of our dequantization mechanism, even with the approximation in Equation 3.

## 4.3 Ablation Studies

Table 3: Watermark detectability (Z-score) under perturbations with *ideal dequantization*. The higher the better for all. **Bold** denotes best, underlined denotes second-best.

| Dataset | Watermark | | Isomorphism | Edge Deletion | | | Edge Addition | | | Node Deletion | | |
|---|---|---|---|---|---|---|---|---|---|---|---|---|
| | | | | 5% | 10% | 20% | 5% | 10% | 20% | 5% | 10% | 20% |
| Planar | Gaussian Shading | (41.6) | 0.4 | 44.3 | 41.8 | 36.5 | 45.4 | 40.0 | 37.9 | 42.0 | 40.2 | 40.9 |
| | TreeRing | (1.2) | 0.0 | 1.0 | 0.8 | 0.4 | 1.1 | 1.0 | 0.7 | 0.8 | 0.5 | 0.0 |
| | Bipartite | (4820.8) | **4842.1** | **4790.6** | **4797.1** | **4424.3** | **3547.2** | **2052.4** | **858.0** | **4833.1** | **4866.8** | **3837.3** |
| | **CheckWate (ours)** | (46.5) | 46.5 | 45.6 | 48.7 | 52.8 | **45.4** | 37.7 | 20.8 | 38.1 | 37.3 | 37.7 |
| Tree | Gaussian Shading | (40.0) | 0.3 | 36.9 | 37.7 | 37.7 | 39.3 | 39.8 | 42.1 | 39.4 | 39.9 | 40.8 |
| | TreeRing | (1.0) | 0.0 | 0.7 | 0.7 | 0.5 | 1.0 | 0.9 | 0.9 | 0.7 | 0.5 | 0.2 |
| | Bipartite | (2661.7) | **2651.3** | **2683.4** | **2788.7** | **2697.1** | **2759.1** | **2741.8** | **2637.4** | **2642.4** | **2636.8** | **2718.7** |
| | **CheckWate (ours)** | (33.2) | 32.3 | 29.5 | 27.2 | 32.1 | 35.1 | 30.2 | 25.6 | 25.0 | 30.0 | 29.3 |
| SBM | Gaussian Shading | (84.4) | 0.3 | 74.1 | 73.8 | 67.1 | 70.4 | 69.0 | 77.1 | 73.7 | 69.9 | 75.4 |
| | TreeRing | (1.0) | 0.0 | 0.7 | 0.4 | 0.0 | 0.3 | 0.0 | 0.0 | 0.3 | 0.0 | 0.0 |
| | Bipartite | (837.1) | **837.1** | **993** | **1230.0** | **964.1** | **15721.4** | **31918.5** | **29797.2** | **944.5** | **7406.3** | **1002.2** |
| | **CheckWate (ours)** | (86.8) | 86.8 | 87.0 | 47.4 | 23.2 | 31.0 | 18.8 | 18.0 | 55.6 | 36.3 | 10.3 |
| Proteins | Gaussian Shading | (119.3) | 11.0 | 150.8 | 151.2 | 161.3 | 150.6 | 153.7 | 152.5 | 152.9 | 153.7 | 153.3 |
| | TreeRing | (1.0) | 0.0 | 1.5 | 1.4 | 1.3 | 1.5 | 1.3 | 1.1 | 1.5 | 1.3 | 1.1 |
| | Bipartite | (1724.0) | **1729.8** | **1643.5** | **1647.3** | **1641.5** | **1652.6** | **1648.4** | **1641.3** | **1647.0** | **1645.3** | **1641.4** |
| | **CheckWate (ours)** | (405.2) | 405.0 | 406.9 | 290.0 | 200.1 | 337.8 | 146.0 | 93.9 | 370.9 | 144.0 | 89.0 |

Table 4: Watermark detectability (Z-score) under perturbations *without sparsification mechanism*. The higher the better for all. **Bold** denotes best, underlined denotes second-best.

| Dataset | Watermark | | Isomorphism | Edge Deletion | | | Edge Addition | | | Node Deletion | | |
|---|---|---|---|---|---|---|---|---|---|---|---|---|
| | | | | 5% | 10% | 20% | 5% | 10% | 20% | 5% | 10% | 20% |
| Planar | Gaussian Shading | (32.0) | 0.0 | 3.9 | 3.0 | 2.7 | 4.7 | 3.0 | **2.9** | 3.6 | 2.8 | 2.0 |
| | TreeRing | (0.0) | 0.0 | 0.0 | 0.0 | 0.0 | 0.0 | 0.0 | 0.0 | 0.0 | 0.0 | 0.0 |
| | Bipartite | (3527.4) | **132.1** | **54.8** | **60.0** | **75.4** | **18.4** | **6.6** | 0.7 | **55.5** | **54.8** | **58.9** |
| | **CheckWate (ours)** | (64.1) | 35.0 | 37.4 | 35.2 | 32.4 | 0.0 | 0.0 | 0.0 | 32.2 | 29.9 | 29.1 |
| Tree | Gaussian Shading | (28.3) | 0.5 | 2.4 | 2.0 | 1.6 | 2.2 | 1.7 | **2.4** | 1.6 | 1.7 | 0.8 |
| | TreeRing | (0.0) | 0.0 | 0.0 | 0.0 | 0.0 | 0.0 | 0.0 | 0.0 | 0.0 | 0.0 | 0.0 |
| | Bipartite | (411.8) | 5.5 | 5.0 | 3.4 | 3.8 | **4.1** | **3.3** | 2.3 | 4.1 | 4.6 | 3.8 |
| | **CheckWate (ours)** | (49.9) | **46.8** | **40.0** | **30.1** | **33.4** | 0.7 | 0.0 | 0.0 | **35.7** | **36.1** | **27.1** |
| SBM | Gaussian Shading | (65.1) | 0.0 | 6.2 | 4.5 | 3.2 | 3.7 | 3.0 | 2.5 | 4.6 | 3.2 | 2.5 |
| | TreeRing | (0.0) | 0.0 | 0.0 | 0.0 | 0.0 | 0.0 | 0.0 | 0.0 | 0.0 | 0.0 | 0.0 |
| | Bipartite | (737200.7) | **127821.4** | **3150.6** | **2302.1** | **395.0** | **49141.5** | 0.0 | 0.0 | **175934** | **2985.9** | **14.2** |
| | **CheckWate (ours)** | (86.8) | 61.2 | 53.4 | 5.5 | 13.6 | 2.3 | 0.3 | 0.0 | 7.3 | 5.6 | 0.7 |
| Proteins | Gaussian Shading | (71.1) | 0.2 | 0.5 | 0.4 | 0.1 | 0.4 | 0.6 | 0.3 | 0.4 | 0.3 | 0.0 |
| | TreeRing | (0.0) | 0.0 | 0.0 | 0.0 | 0.0 | 0.0 | 0.0 | 0.0 | 0.0 | 0.0 | 0.0 |
| | Bipartite | (97.8) | 97.3 | 97.3 | 97.3 | 97.3 | **97.3** | **97.3** | **97.3** | 97.3 | **97.3** | **97.3** |
| | **CheckWate (ours)** | (410.9) | **221.7** | **239.7** | **136.4** | **109.7** | 3.1 | 1.6 | 0.5 | **208.9** | 20.1 | 26.4 |

**Ideal dequantization** We perform experiments in which we assume the permutation matrix $\mathbf{P}$ to be known. We dequantize $\mathbf{G}^{0'*} = \mathbf{A}' - \mathbf{P}\left(\mathbf{A} - \mathbf{G}^0\right)\mathbf{P}^{-1}$ and consider $\mathbf{G}^{0'*}$ to be the ideal result of dequantization. Table 3 showcases the results. This allows us to analyze the difference in performance given by the approximation in Equation 3. `CheckWate` achieves enhanced watermark strength, especially under heavier attacks. This can be explained as the eigenvectors of $\mathbf{G}^T$ tend to change more when under heavier perturbations, making the reconstruction from Equation 3 less robust. Furthermore, we emphasize that isomorphism does not degrade `CheckWate` detectability, proving graph-invariance of the checkerboard watermarking. We further see that the baseline that is affected the most is Gaussian Shading. Albeit, it continues to have insufficient detectability when under isomorphism as its watermark is not graph-invariant.

**Robust detection mechanism** Finally, we perform experiments with the robust detection mechanism disabled. Results are showcased in Table 4. We can clearly see that most Z-scores are reduced. Notably, under the edge addition attack, `CheckWate` is not detectable even under lighter attacks. This happens as the reconstructed latent diverges from the GOE assumption. This leads to the eigenvalues diverging from the bulk. At the same time, applying the robust detection mechanism constrains the latent to the sparsified GOE, making its behavior predictable, and forcing the eigenvalues to stay within the bulk. We showcase a qualitative comparison in Fig. 8 from Appendix H.

## 5 CONCLUSIONS

Motivated by the need to efficiently verifying ownership of synthetic graph data, we propose `CheckWate`, the first sampling-time watermark for graph diffusion models. `CheckWate` embeds a checkerboard pattern in the noisy latent and detects the watermark in polynomial time using the noisy latent eigenvalues. The novel design of `CheckWate` leverages random matrix theory to solve multiple hard graph watermarking challenges: bypassing NP-hardness of verifying graphs arising from graph isomorphism, and dequantizing discrete and isomorphic graph representations. Our watermark is not only theoretically grounded in watermark verification time and graph reconstruction error, but also practically robust against graph-modifications. Our evaluation across four datasets shows that `CheckWate` achieves state-of-the-art generative quality and remains detectable under graph-specific attacks such as isomorphism, while watermarks of prior art are barely detectable. We discuss limitations and future work in Appendix E.

### REPRODUCIBILITY STATEMENT

To ensure the reproducibility of our research, we include: code of the proposed framework, pseudocode of watermark detection in appendix. All used datasets are publicly available and instructions to reproduce our results are provided in the code repository.

### ETHICS STATEMENT

Our proposed graph watermarking framework has broad applications in claiming ownership over synthetic network data such as molecular structures used in drug discovery or material science applications and human interactions on social media, or professional networks.

This paper was written with the aid of publicly available LLMs in tasks such as grammar checking, spelling correction, and minor rephrasing.

### ACKNOWLEDGMENTS

This work is funded by NExTWORKx, a collaboration between TU Delft and KPN on future telecommunication networks. This research was partly funded by the DYMAN project funded by the European Union - European Innovation Council under G.A. n. 101161930. M. Kitsak has been supported by the Dutch Research Council (NWO) grants OCENW.M20.244 and VI.C.242.106.

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

## LIST OF SYMBOLS

$\delta(\cdot)$     Probability density function of Dirac distribution

$\Lambda$     The diagonal matrix of the eigenvalues, i.e., $\text{diag}(\lambda_1, \ldots, \lambda_N)$

$\lambda$     Eigenvalue

$\mathbf{A}$     Binary graph adjacency matrix

$\mathbf{A}'$     Binary graph adjacency matrix to verify

$\mathbf{G}^0$     Generated graph (in the continuous space)

$\mathbf{G}^T$     Noisy latent

$\mathbf{G}^{T'}$     Reconstructed noisy latent

$\mathbf{V_x}$     Eigenvectors of matrix $\mathbf{X}$

$\mathbf{X}_N$     A normalized matrix, i.e., $\mathbf{X}_N = \mathbf{X}/\sqrt{N}$

$\mathbf{X}_{ij}$     Entry $i, j$ of matrix $\mathbf{X}$

$\mathcal{N}(\mu, \sigma)$     Gaussian distribution with mean $\mu$ and std $\sigma$

$\phi(\cdot)$     Probability density function of Gaussian distribution

$\rho(\cdot)$     Probability density function of eigenvalues of a normalized matrix

$\text{diag}(\cdot)$     Diagonal matrix

$e(\lambda)$     Algebraic multiplicity of eigenvalue $\lambda$

## A    BACKGROUND ON DIFFUSION MODELS

**DDPM** Diffusion models generate data starting from a noisy latent representation. *Denoising Diffusion Probabilistic Model* (DDPM) (Ho et al., 2020) has been at the forefront of generation of synthetic data. This framework aims at transitioning from a latent sampled from noise distribution ($z_T \sim \mathcal{N}(0,1)$) to a sample of the data distribution $z_0$ through a iterative process. More precisely, at each step $t$, a neural network $\epsilon_\theta$ predicts the noise $\epsilon_\theta(t, z_t)$ to predict the next sample $z_{t-1}$ as:

$$z_{t-1} = \sqrt{\alpha_{t-1}} \left( \frac{z_t - \sqrt{1 - \alpha_t}\epsilon_\theta(t, z_t)}{\sqrt{\alpha_t}} \right) + \sqrt{1 - \alpha_{t-1} - \sigma_t^2} \cdot \epsilon_\theta(t, z_t) + \sigma_t \epsilon_t \qquad (7)$$

where $\alpha_1, \ldots, \alpha_T$ are computed from a predefinied variance schedule. $\epsilon_t \sim \mathcal{N}(0,1)$ is independent standard Gaussian noise. $\sigma_t$ is noise that yields diversification in the generative process.

**DDIM** By setting $\sigma_t = 0$ the generative process becomes deterministic, i.e., implicit. Meaning that, for a starting noise $z_T$, the generative process deterministically yields to the same $z_0$. Notably, if the size of the steps is small, i.e., large $T$, the generative process can be accurately reversed and $z_T$ can be reconstructed from $z_0$ via:

$$z_{t+1} = \sqrt{\frac{\alpha_{t+1}}{\alpha_t}} z_t + \left( \sqrt{1 + \alpha_{t+1}} - \sqrt{\frac{\alpha_{t+1}}{\alpha_t} - \alpha_{t+1}} \right) \epsilon_\theta(t, z_t) \qquad (8)$$

based on the approximation $\epsilon_\theta(t, z_t) \approx \epsilon_\theta(t - 1, z_{t-1})$. This paradigm is called the *Denoising Diffusion Implicit Model* (DDIM) (Song et al., 2020). Due to its capability of reconstructing noisy latent, it has been used as the backbone of multiple watermarking methodologies (Wen et al., 2023; Yang et al., 2024; Zhu et al., 2025; Soi et al., 2025).

**DDBM** *Denoising Diffusion Bridge Models* (DDBM) (Zhou et al., 2023) generalize diffusion by allowing the endpoints of the diffusion process to be arbitrary distributions rather than always starting from random noise. This enables denoising diffusion to operate on usecases such as image editing. DDBM learn a bridge score function $s(t, x)$ via a neural network. Then, the *reverse process* can be expressed as a stochastic differential equation (SDE):

$$d\mathbf{x}^t = [f(\mathbf{x}^t, t) - g(t)^2 s(\mathbf{x}^t, t)]dt + g(t)d\mathbf{W}^t \qquad (9)$$

where $f(x, t)$ and $g(t)^2$ come from the forward reference diffusion process, and $\mathbf{W}^t$ is a standard Wiener process that introduces diversification in the generative process.

### A.1    GRAPH DIFFUSION MODELS

**Discrete Graph Diffusion** Some of the state-of-the-art diffusion models perform diffusion using discrete graph representations in the latent space. DiGress (Vignac et al., 2022) is one of the most popular examples. DiGress represents noisy latents as Erdős-Renyi (ER) random graphs in which each edge is independently sampled with $50\%$ chance. The generation follows a discrete denoising diffusion process inspired by DDPM: at each timestep $t$, a neural network predicts a distribution $p(\mathbf{G}^{t-1}|\mathbf{G}^t)$, from which the next graph $\mathbf{G}^{t-1}$ is sampled.

**Continuous Graph Diffusion** Other approaches operate on continuous graph representations. GruM (Jo et al., 2023), for instance, models noisy latents as Gaussian Orthogonal Ensembles. Then, the diffusion process is modeled through DDBM process that moves toward the target data distribution. At each step, $\mathbf{G}^t$ represents the mixture of the possible generated graphs. Relying on the Ornstein-Ulhenback bridge process, GruM is shown to converge to the data distribution up to quantization.

## B    BASELINES

**Gaussian Shading** (Yang et al., 2024) is a sampling-time watermarking framework designed for images. The watermark is embedded in the latent using a secure stream cipher such as ChaCha20 (Bernstein) to get a uniformly distributed random bits. The noisy latent is generated from these bits using distribution preserving sampling. When the noisy latent is reconstructed, the bits are reconstructed. Then, the author can prove their ownership by comparing them with the

ones generated by the secure stream cipher. Gaussian Shading is proven to deliver lossless generative performance but is not isomorphism invariant as the pattern designed by the stream cipher is disrupted.

**TreeRing** (Wen et al., 2023) is a sampling-time watermarking framework designed for images. TreeRing embeds the watermark in the Fourier space of the noisy latent. When the noisy latent is reconstructed, the watermark is extracted via the Fourier transform. Then, its presence is detected via L1 similarity from the original watermarking key and the reconstructed one. TreeRing yields reduced generative performance as it heavily disrupts the initial latent from the Gaussian assumption. Furthermore, it is not isomorphism invariant as its pattern is disrupted.

**Bipartite** is a graph-invariant baselines we developed based on bipartite graphs. We leverage the fact that graph bipartivity can be verified regardless of the graph representation and that it is a monotonic property. Meaning, a subgraph of a bipartite graph is still bipartite. It starts by generating a complete bipartite graph of size $N \times N$. From its adjacency matrix, it performs distribution preserving sampling to generate the noisy latent. To detect the watermark, the noisy latent is reconstructed. Then, we discretize the noisy latent as:

$$L = \begin{cases} 0 & \text{if } \Phi(L) < 0.5 \\ 1 & \text{otherwise} \end{cases}$$

Where $\Phi$ is the CDF of the standard normal distribution. To increase its robustness, we leverage monotonicity to sample a subgraph of L

$$L = \begin{cases} 0 & \text{if } \Phi(L) < 0.75 \\ 1 & \text{otherwise} \end{cases}$$

Recall that monotonicity means that a subgraph of a bipartite graph is always bipartite. The score of the watermark is computed via the spectral bipartivity $\beta(z)$ (Estrada & Rodríguez-Velázquez, 2005). In general $\frac{1}{2} < \beta(z) \leq 1$. With $\beta(z) = 1$ if and only if a graph is bipartite, and $\beta(z) \to \frac{1}{2}$ when $z$ is the complete graph and $N \to \infty$. Bipartite leads to reduced generative performance as it generates noisy latents of non-independent Gaussian noise. It is isomorphism invariant as bipartivity is a graph property (Estrada & Rodríguez-Velázquez, 2005).

## C  PROOFS

### C.1  PROOF OF THEOREM 3.1

From Chen et al. (2020) we know that the largest $N - k$ eigenvalues are of magnitude: $\sqrt{N}W_u/k + \mathcal{O}(1)$. Hence,

$$\mathbb{E}_{\mathbf{G}^W \sim (k,W)\text{-CBE}} \left[ \frac{\sum_{i=1}^{N-k} \lambda_i \left( \mathbf{G}^W \right)}{(N-k)} \right] = \left( \sqrt{N}w/k + \mathcal{O}(1) \right) \tag{10}$$

We now derive the expected average magnitude of the largest $N - k$ eigenvalues of the GOE. For the sake of simplifying notation, let $k' = \frac{k}{N} \in [0,1]$

$$\mathbb{E}_{\mathbf{G}^{NW} \sim \text{GOE}} \left[ \frac{\sum_{i=1}^{N-k} \lambda_i \left( \mathbf{G}^{NW} \right)}{(N-k)} \right] = \mathbb{E}\left[ PDF_{Wigner}(x) \mid 2k' \leq x \leq 2 \right] \tag{11}$$

$$= \frac{\int_{2k'}^{2} x PDF_{Wigner}(x)dx}{\int_{2k'}^{2} PDF_{Wigner}(x)dx} \tag{12}$$

$$= \frac{\int_{2k'}^{2} x \frac{\sqrt{4-x^2}}{2\pi} dx}{CDF_{Wigner}(2) - CDF_{Wigner}(k')} \tag{13}$$

$$= \frac{\left(4 - 4k'^2\right)^{3/2}}{6\pi \left[ \frac{1}{2} - \frac{2k'\sqrt{4-4k'^2}}{4\pi} - \frac{\arcsin\left(\frac{2k'}{2}\right)}{\pi} \right]} \tag{14}$$

$$\approx \mathcal{O}(k'^2) = \mathcal{O}(k^2) \tag{15}$$

Finally, we can derive the watermark detectability of `CheckWate`.

$$\mathbb{E}\left[\sum_{i=1}^{N-k} \lambda_i(G^W) - \lambda_i(G^{NW})\right] = \mathbb{E}\left[\sum_{i=1}^{N-k} \lambda_i(G^W)\right] - \mathbb{E}\left[\sum_{i=1}^{N-k} \lambda_i(G^{NW})\right] \tag{16}$$

$$= \sqrt{N}w/k + \mathcal{O}(1) - \mathcal{O}(k^2) \tag{17}$$

### C.2 PROOF OF THEOREM 3.2

With $\mathbf{G}^{0'}$ defined as in Section 3.3.

$$\mathbf{G}^{0'} = \mathbf{P}\mathbf{G}^0\mathbf{P}^{-1} \tag{18}$$

$$= (\mathbf{P}\mathbf{V}_A)\mathbf{V}_{\mathbf{A}}^{-1}\mathbf{G}^0\mathbf{V}_{\mathbf{A}}(\mathbf{P}\mathbf{V}_{\mathbf{A}})^{-1} \tag{19}$$

$$\approx (\mathbf{P}\mathbf{V}_{\mathbf{A}}\mathbf{Q})\mathbf{V}_{\mathbf{A}}^{-1}\mathbf{G}^0\mathbf{V}_{\mathbf{A}}(\mathbf{P}\mathbf{V}_{\mathbf{A}}\mathbf{Q})^{-1} \tag{20}$$

$$= \mathbf{V}_{\mathbf{A}'}\mathbf{V}_{\mathbf{A}}^{-1}\mathbf{G}^0\mathbf{V}_{\mathbf{A}}\mathbf{V}_{\mathbf{A}'}^{-1} \tag{21}$$

Equality for Equation 20 holds whenever $\mathbf{Q} = \mathbf{I}$.

### C.3 PROOF OF THEOREM 3.3

Whenever $\mathbf{A}$ has eigenvalue multiplicity $> 1$, the reconstruction error can be quantified as:

$$\|\mathbf{G}^{0'} - \mathbf{G}^0\|_F = \|\mathbf{V}_{\mathbf{A}}\mathbf{Q}\mathbf{V}_{\mathbf{A}}^{-1}\mathbf{G}^0\mathbf{V}_{\mathbf{A}}\mathbf{Q}^{-1}\mathbf{V}_{\mathbf{A}}^{-1} - \mathbf{G}^0\|_F \tag{22}$$

$$= \|\mathbf{V}_{\mathbf{A}}^{-1}(\mathbf{V}_{\mathbf{A}}\mathbf{Q}\mathbf{V}_{\mathbf{A}}^{-1}\mathbf{G}^0\mathbf{V}_{\mathbf{A}}\mathbf{Q}^{-1}\mathbf{V}_{\mathbf{A}}^{-1} - \mathbf{G}^0)\mathbf{V}_{\mathbf{A}}\|_F \tag{23}$$

$$= \|\mathbf{Q}\mathbf{V}_{\mathbf{A}}^{-1}\mathbf{G}^0\mathbf{V}_{\mathbf{A}}\mathbf{Q}^{-1} - \mathbf{V}_{\mathbf{A}}^{-1}\mathbf{G}^0\mathbf{V}_{\mathbf{A}}\|_F \tag{24}$$

$$= \sum_{r,s=1}^{m} \left(\|\mathbf{Q}_r\mathbf{G}'_{r,s}\mathbf{Q}_s^{-1} - \mathbf{G}'_{r,s}\|_F^2\right)^{1/2} \tag{25}$$

where $\|\cdot\|_F$ is the Frobenius norm.

## D PSEUDOCODE

---

**Algorithm 1** `CheckWate` Detection

---

**Input**: target graph $\mathbf{A}'$, reference graph $\mathbf{G}^0$, denoising model $\mathcal{M}$
1: $\mathbf{V}_{\mathbf{A}'} \leftarrow \text{Eigenvectors}(\mathbf{A}')$
2: $\mathbf{V}_{\mathbf{A}} \leftarrow \text{Eigenvectors}(\text{Quantize}(\mathbf{G}^0))$
3: $\mathbf{G}^{0'} \leftarrow \mathbf{V}_{\mathbf{A}'}\mathbf{V}_{\mathbf{A}}^{-1}\mathbf{G}^0\mathbf{V}_{\mathbf{A}}\mathbf{V}_{\mathbf{A}'}^{-1}$                                    {Equation 3}
4: **for** $t \leftarrow 1, 2, \dots, T$ **do**
5:     $\mathbf{G}^{t'} \leftarrow \mathcal{M}^{-1}\left(\mathbf{G}^{(t-1)'}\right)$                                    {Reverse DBIM}
6: **end for**
7: **for** $(i, j) \in N \times N$ **do**
8:     **if** $\max\left(\phi(\mathbf{G}_{ij}^{T'}), \delta(\mathbf{G}_{ij}^{T'})\right) \leq \theta$ **then**                                    {Equation 5}
9:         $\mathbf{G}_{ij}^{T'} \leftarrow 0$
10:     **end if**
11: **end for**
12: $\mathbf{G}_N^{T'} = \frac{\mathbf{G}^{T'}}{\sqrt{N}}$                                    {Normalization}
13: $\text{BlipEigenvalues} \leftarrow \left[\lambda \mid \lambda \in \text{Eigenvalues}\left(\mathbf{G}^{T'}\right), |\lambda| \gg 2\right]$
14: **if** $|\text{BlipEigenvalues}| \geq N - k$ **then**
15:     **Return** *SUCCESS*
16: **end if**
17: **Return** *FAIL*

---

# E LIMITATIONS

**Watermark on discrete latents** Some graph diffusion models such as DiGress (Vignac et al., 2022) rely on discrete noisy latent and discrete denoising steps of DDPM. While inverting discrete denoising diffusion remains an untackled problem, techniques similar to the checkerboard watermark of `CheckWate` can also be applied to discrete noisy latents. The key idea of CheckWate watermark lies in moving some of the eigenvalues outside of the bulk (i.e., Wigner semicircle) while applying minimal changes to the latent. In the context of continuous latents, we achieve this by applying the checkered entries. We believe that the same idea can be applied in discrete space in the following way.

Eigenvalues of discrete noisy latents (i.e., Erdos-Renyi matrices) also follow the Wigner semicircle law, similarly to GOEs. We are interested in moving some of these eigenvalues outside the bulk regime. Budel & Van Mieghem (2021) studied the relationship between presence of communities and eigenvalues in the blip. We suggest that enforcing the presence of communities in parts of the noisy latent can be used to reproduce `CheckWate` behavior in the discrete scenario. This can be leveraged to extend the checkered watermark behavior to discrete noisy latents. Furthermore, sparsifying ER graphs, reduces the magintude of its eigenvalues, similarly to the sparsified GOE. Thus, even the robust detection mechanism `CheckWate` remains applicable.

**Non-Blind Watermark** Our watermark is non-blind, meaning that the original data is needed to verify the presence of the watermark. State-of-the-art watermarking methodologies for images, tabular data, and time series provide blind watermarking, meaning that the watermark can be extracted and verified even without the generated data. No watermarking framework for graph data currently supports blind watermarking. Addressing computational feasibility of non-blind graph watermarking is a key step to enable future development in this field. Extending `CheckWate` to further support blindness is an interesting research gap that will be addressed by future work.

**Provable resistance to forgery** `CheckWate` resistance to forgery relies on hardness of dequantization. Despite our experiments in Table 1 show that not using the key leads to impossible verification, it is not possible to prove forgery resistance by applying the key after the diffusion process.

# F ABLATION STUDY ON DIFFERENT VALUES OF $k$, $W$

Here, we widen our results with an extensive analysis of how $k$ and $W$ can affect Z-score and generative quality. The key observations are, as expected, generation quality degrades for larger values $W$ of and smaller $k$. Accordingly, Z-score increases when $k$ is smaller and $W$ larger. This confirms the theory discussed in Section 3.2.

# G ROC CURVES

We report the ROC curves of the experiments from Table 1 and 2. We provide results under Planar and SBM datasets (the latter one being the dataset with the lowest z-scores) with four different attacks at their maximum strength:

All figures show an AUC of 1.0, which means `CheckWate` always manages to achieve 100% True Positive Rate and 0% False Positive Rate.

We further provide the same ROC curves without the latent sparsification mechanism:

From the plots, we can see that under strong edge additions, the AUC of CheckWate reduces severely: 0.5 for Planar and 0.75 for SBM. A result that indicates little to no discriminatory capability.

Furthermore, we would like to highlight figures in Appendix H, in which we perform qualitative analysis on the eigenvalue distributions of our experiments. Especially, Fig. 8 showcases how the eigenvalues of the reconstructed latents tend to explode under edge additions when no latent sparsification is applied.

Table 5: Ablation study on different values for $k, W$ on Planar dataset.

| Parameters | degree | cluster | orbit | spectral | V.U.N. | Z-Score |
|---|---|---|---|---|---|---|
| $k = 0.9, W = \pm1.0$ | 0.0006 | 0.0406 | 0.0039 | 0.0086 | 0.55 | 7.64 |
| $k = 0.9, W = \pm2.0$ | 0.0009 | 0.0446 | 0.0113 | 0.0068 | 0.65 | 38.76 |
| $k = 0.9, W = \pm3.0$ | 0.0007 | 0.053 | 0.0092 | 0.0087 | 0.675 | 95.47 |
| $k = 0.9, W = \pm4.0$ | 0.0011 | 0.0651 | 0.0154 | 0.0096 | 0.525 | 176.88 |
| $k = 0.7, W = \pm1.0$ | 0.0009 | 0.0424 | 0.0022 | 0.008 | 0.325 | 19.35 |
| $k = 0.7, W = \pm2.0$ | 0.0007 | 0.0541 | 0.0119 | 0.0099 | 0.4 | 83.17 |
| $k = 0.7, W = \pm3.0$ | 0.0015 | 0.0629 | 0.0283 | 0.0109 | 0.25 | 193.44 |
| $k = 0.7, W = \pm4.0$ | 0.0021 | 0.0779 | 0.0352 | 0.0111 | 0.175 | 349.78 |
| $k = 0.5, W = \pm1.0$ | 0.0007 | 0.0592 | 0.0046 | 0.0084 | 0.25 | 29.61 |
| $k = 0.5, W = \pm2.0$ | 0.0014 | 0.0664 | 0.0178 | 0.0114 | 0.275 | 126.95 |
| $k = 0.5, W = \pm3.0$ | 0.0021 | 0.0984 | 0.0354 | 0.0118 | 0.125 | 292.39 |
| $k = 0.5, W = \pm4.0$ | 0.0047 | 0.1028 | 0.0631 | 0.0121 | 0.275 | 525.72 |
| $k = 0.2, W = \pm1.0$ | 0.0031 | 0.2282 | 0.0197 | 0.0115 | 0 | 134.64 |
| $k = 0.2, W = \pm2.0$ | 0.007 | 0.259 | 0.0734 | 0.0167 | 0 | 561.77 |
| $k = 0.2, W = \pm3.0$ | 0.0132 | 0.2624 | 0.0435 | 0.0164 | 0 | 1288.26 |
| $k = 0.2, W = \pm4.0$ | 0.0198 | 0.2687 | 0.0729 | 0.0231 | 0 | 2303.01 |

Table 6: Ablation study on different values for $k, W$ on Tree dataset.

| Parameters | degree | cluster | orbit | spectral | V.U.N. | Z-Score |
|---|---|---|---|---|---|---|
| $k = 0.9, W = \pm1.0$ | 0.0001 | 0 | 0 | 0.0093 | 0.55 | 6.07 |
| $k = 0.9, W = \pm2.0$ | 0.0002 | 0 | 0 | 0.0088 | 0.55 | 32.22 |
| $k = 0.9, W = \pm3.0$ | 0.0001 | 0.0001 | 0 | 0.0094 | 0.5 | 81.35 |
| $k = 0.9, W = \pm4.0$ | 0 | 0.0001 | 0 | 0.0093 | 0.65 | 152.06 |
| $k = 0.7, W = \pm1.0$ | 0 | 0.0001 | 0 | 0.0084 | 0.575 | 16.43 |
| $k = 0.7, W = \pm2.0$ | 0.0002 | 0.0001 | 0.0001 | 0.0092 | 0.55 | 73.77 |
| $k = 0.7, W = \pm3.0$ | 0.0002 | 0 | 0.0002 | 0.0089 | 0.35 | 172.15 |
| $k = 0.7, W = \pm4.0$ | 0.0001 | 0 | 0.0002 | 0.0088 | 0.25 | 316.53 |
| $k = 0.5, W = \pm1.0$ | 0.0004 | 0 | 0.0001 | 0.0096 | 0.65 | 25.04 |
| $k = 0.5, W = \pm2.0$ | 0 | 0.0001 | 0.0002 | 0.0117 | 0.55 | 113.12 |
| $k = 0.5, W = \pm3.0$ | 0.0003 | 0.0001 | 0.0003 | 0.0111 | 0.425 | 262.96 |
| $k = 0.5, W = \pm4.0$ | 0.0002 | 0.0005 | 0.0004 | 0.0093 | 0.325 | 471.28 |
| $k = 0.2, W = \pm1.0$ | 0.0001 | 0.1547 | 0.0002 | 0.0146 | 0.05 | 120.53 |
| $k = 0.2, W = \pm2.0$ | 0.0006 | 0.4638 | 0.0005 | 0.0148 | 0 | 514.98 |
| $k = 0.2, W = \pm3.0$ | 0.0007 | 0.5882 | 0.0009 | 0.0182 | 0 | 1162.53 |
| $k = 0.2, W = \pm4.0$ | 0.0008 | 0.6223 | 0.0015 | 0.0151 | 0 | 2068.21 |

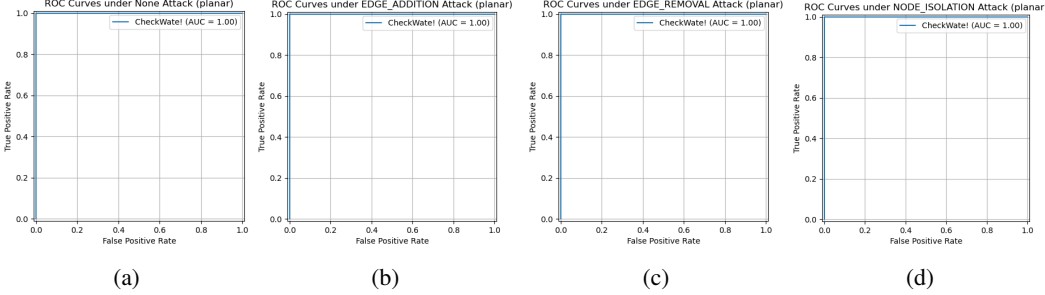

(a)  (b)  (c)  (d)

Figure 4: ROC curves of `CheckWate` on Planar dataset with latent sparsification. (a) ROC curve under no attack; (b) ROC curve under edge addition 20%; (c) ROC curve under edge removal 20%; (d) ROC curve under node deletion 20%. AUC is 1.0, indicating perfect discriminatory capability.

Table 7: Ablation study on different values for $k, W$ on SBM dataset.

| Parameters | degree | cluster | orbit | spectral | V.U.N. | Z-Score |
|---|---|---|---|---|---|---|
| $k = 0.9, W = \pm 1.0$ | 0.0025 | 0.0488 | 0.0507 | 0.0064 | 0.775 | 5.67 |
| $k = 0.9, W = \pm 2.0$ | 0.0021 | 0.0511 | 0.0574 | 0.0045 | 0.6 | 27.44 |
| $k = 0.9, W = \pm 3.0$ | 0.0026 | 0.052 | 0.0676 | 0.0056 | 0.625 | 64.8 |
| $k = 0.9, W = \pm 4.0$ | 0.0022 | 0.0514 | 0.0665 | 0.0056 | 0.6 | 117.22 |
| $k = 0.7, W = \pm 1.0$ | 0.0035 | 0.0501 | 0.072 | 0.0067 | 0.725 | 13.26 |
| $k = 0.7, W = \pm 2.0$ | 0.0048 | 0.0533 | 0.0517 | 0.0055 | 0.575 | 55.14 |
| $k = 0.7, W = \pm 3.0$ | 0.004 | 0.055 | 0.075 | 0.0055 | 0.475 | 126.23 |
| $k = 0.7, W = \pm 4.0$ | 0.0044 | 0.057 | 0.0855 | 0.0057 | 0.525 | 225.89 |
| $k = 0.5, W = \pm 1.0$ | 0.0024 | 0.0524 | 0.0825 | 0.0049 | 0.775 | 20.29 |
| $k = 0.5, W = \pm 2.0$ | 0.0044 | 0.0528 | 0.0829 | 0.0046 | 0.75 | 83.1 |
| $k = 0.5, W = \pm 3.0$ | 0.0048 | 0.0576 | 0.0754 | 0.0058 | 0.625 | 188.22 |
| $k = 0.5, W = \pm 4.0$ | 0.0048 | 0.058 | 0.0916 | 0.0055 | 0.6 | 336.24 |
| $k = 0.2, W = \pm 1.0$ | 0.0029 | 0.0514 | 0.0783 | 0.0047 | 0.725 | 82.68 |
| $k = 0.2, W = \pm 2.0$ | 0.0036 | 0.0507 | 0.0693 | 0.0063 | 0.575 | 339.44 |
| $k = 0.2, W = \pm 3.0$ | 0.0033 | 0.0497 | 0.0485 | 0.0057 | 0.425 | 769.34 |
| $k = 0.2, W = \pm 4.0$ | 0.0024 | 0.0502 | 0.0571 | 0.0065 | 0.3 | 1373.52 |

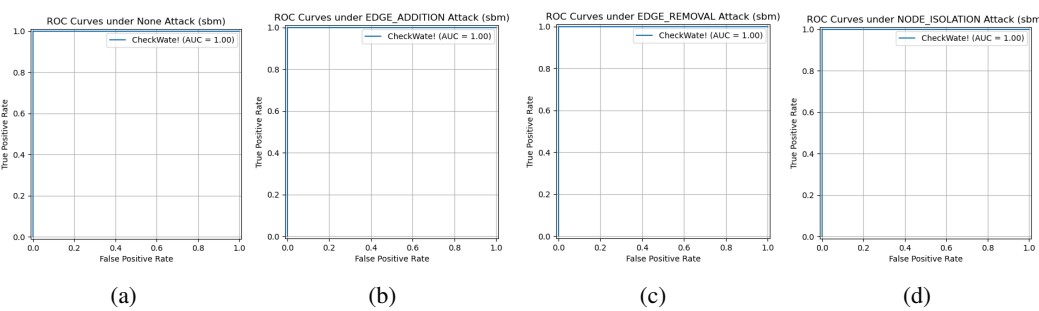

Figure 5: ROC curves of `CheckWate` on SBM dataset with latent sparsification. (a) ROC curve under no attack; (b) ROC curve under edge addition 20%; (c) ROC curve under edge removal 20%; (d) ROC curve under node deletion 20%. AUC is 1.0, indicating perfect discriminatory capability.

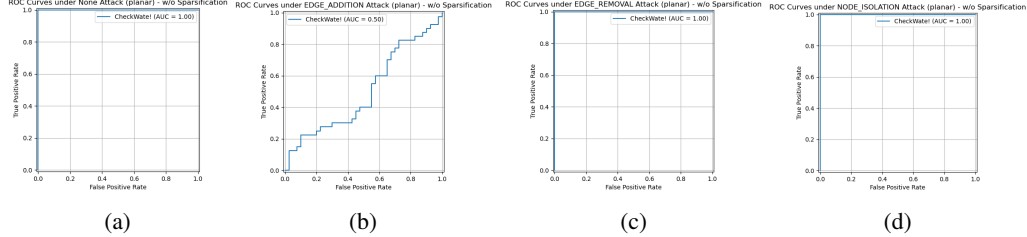

Figure 6: ROC curves of `CheckWate` on Planar dataset without latent sparsification. (a) ROC curve under no attack; (b) ROC curve under edge addition 20%; (c) ROC curve under edge removal 20%; (d) ROC curve under node deletion 20%. Under edge addition, AUC is 0.5, indicating limited discriminatory capability.

Altogether, these results show that: (i) `CheckWate` provides remarkable performance under TPR and FPR; (ii) Latent sparsification is key for preventing performance degradation under major perturbations.

## H    QUALITATIVE RESULTS

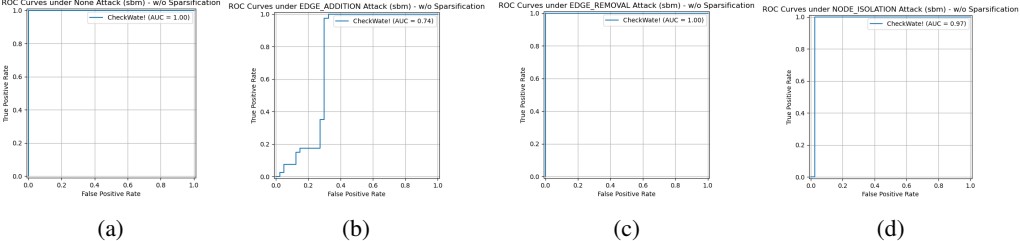

Figure 7: ROC curves of `CheckWate` on SBM dataset without latent sparsification. (a) ROC curve under no attack (b) ROC curve under edge addition 20% (c) ROC curve under edge removal 20% (d) ROC curve under node deletion 20%. Under edge addition, AUC is 0.74, indicating limited discriminatory capability.

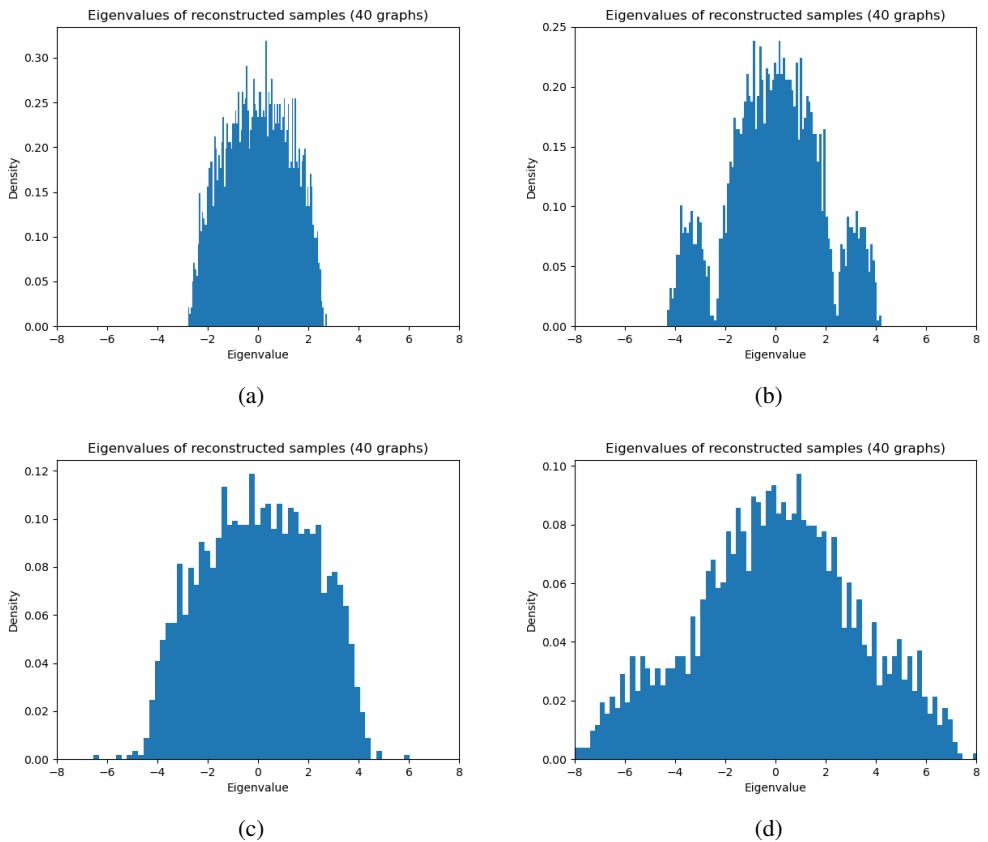

Figure 8: Comparison of eigenvalues with and without anomaly detection mechanism on Planar dataset under edge addition attack (10%). (a) and (b) show reconstructed eigenvalues of *No watermark* and `CheckWate` respectively, with anomaly detection. (c) and (d) show reconstructed eigenvalues of *No watermark* and `CheckWate` respectively, without anomaly detection.

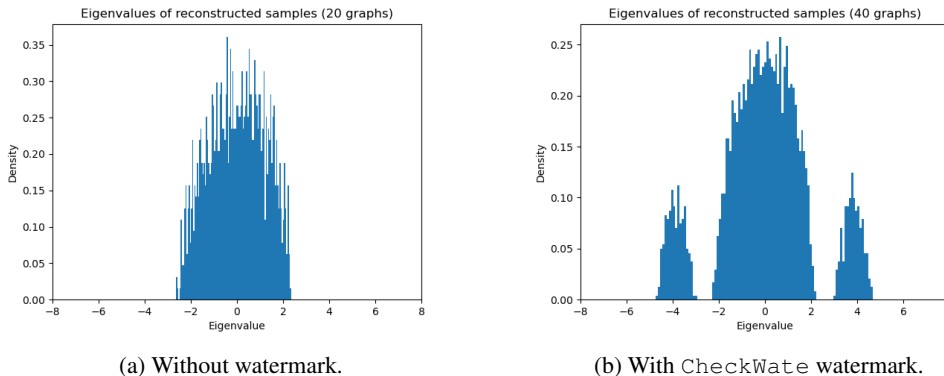

(a) Without watermark.

(b) With `CheckWate` watermark.

Figure 9: Distribution of reconstructed eigenvalues on Planar dataset under no attack.

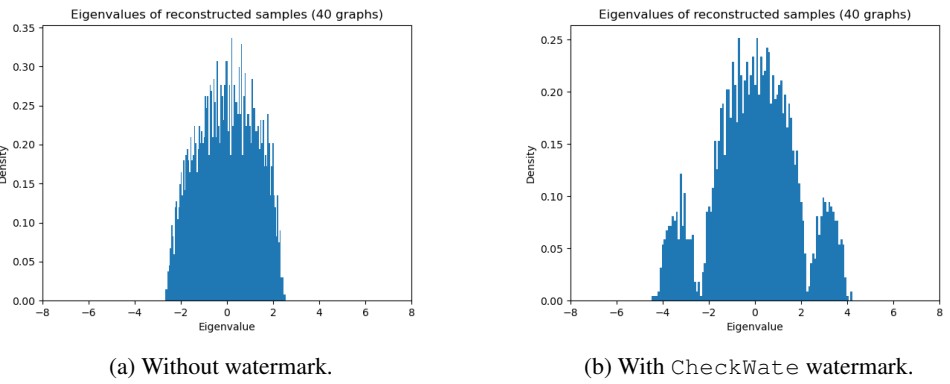

(a) Without watermark.

(b) With `CheckWate` watermark.

Figure 10: Distribution of reconstructed eigenvalues on Planar dataset under isomorphism.

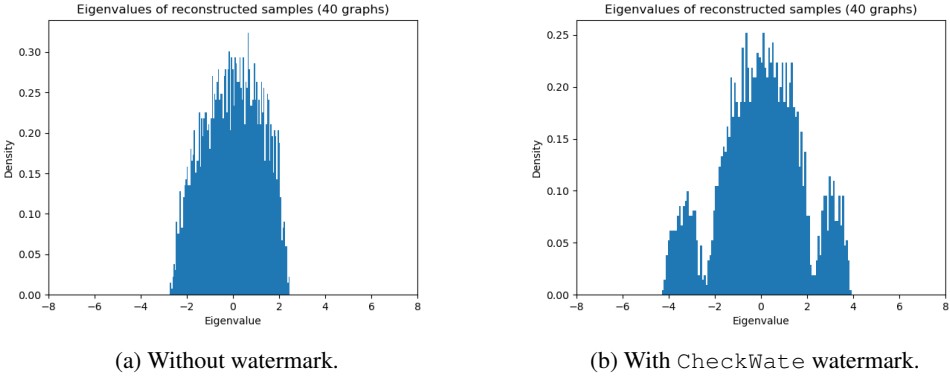

(a) Without watermark.

(b) With `CheckWate` watermark.

Figure 11: Distribution of reconstructed eigenvalues on Planar dataset under 20% edge removal.

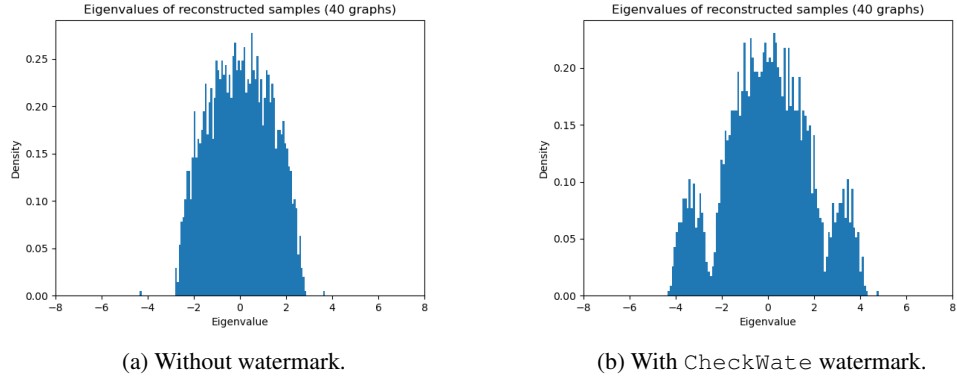

(a) Without watermark.

(b) With `CheckWate` watermark.

Figure 12: Distribution of reconstructed eigenvalues on Planar dataset under 20% edge addition.

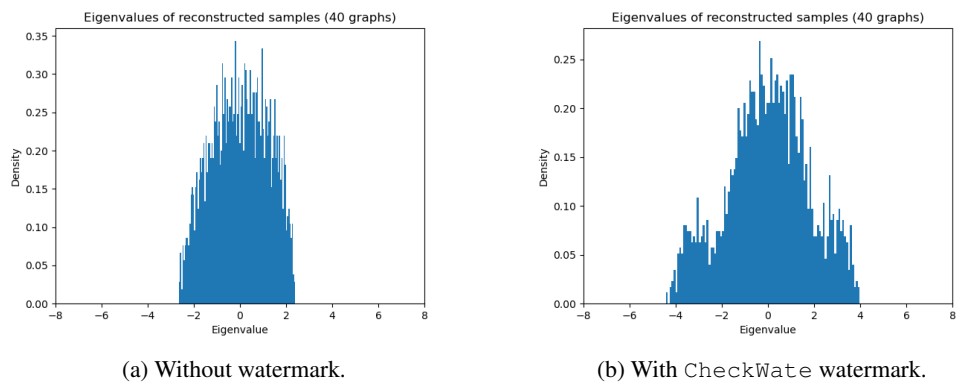

(a) Without watermark.

(b) With `CheckWate` watermark.

Figure 13: Distribution of reconstructed eigenvalues on Planar dataset under 20% node deletion.

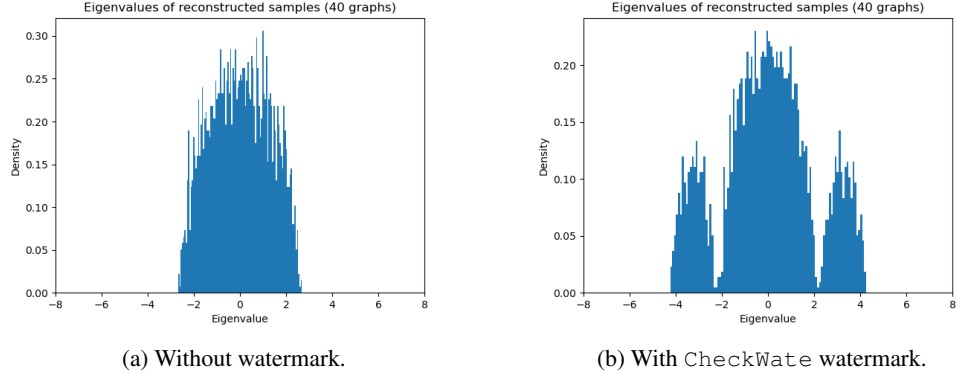

(a) Without watermark.

(b) With `CheckWate` watermark.

Figure 14: Distribution of reconstructed eigenvalues on Tree dataset under no attack.

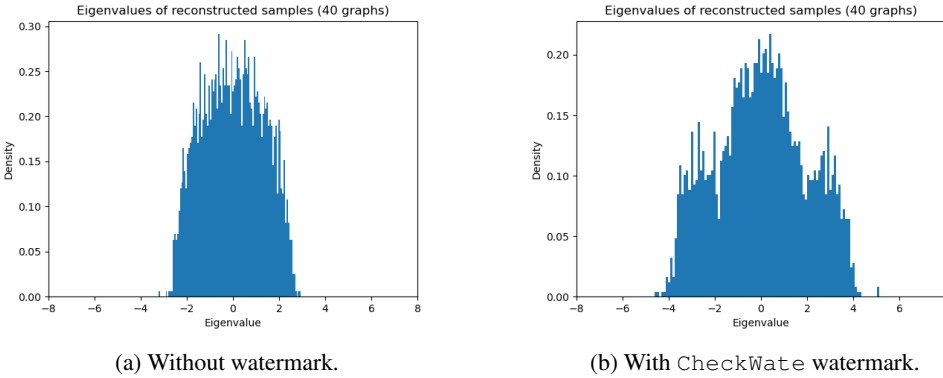

(a) Without watermark.

(b) With `CheckWate` watermark.

Figure 15: Distribution of reconstructed eigenvalues on Tree dataset under isomorphism.

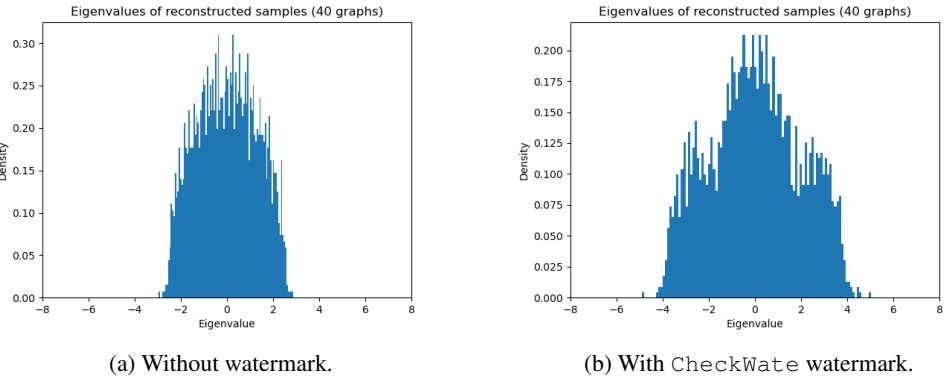

(a) Without watermark.

(b) With `CheckWate` watermark.

Figure 16: Distribution of reconstructed eigenvalues on Tree dataset under 20% edge removal.

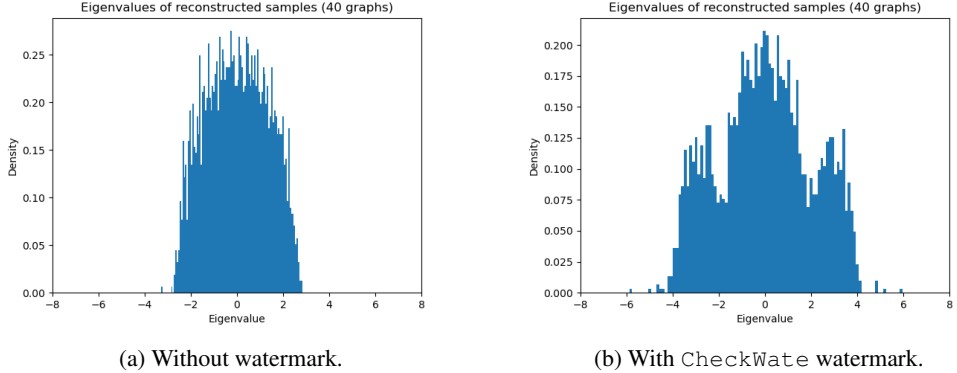

(a) Without watermark.

(b) With `CheckWate` watermark.

Figure 17: Distribution of reconstructed eigenvalues on Tree dataset under 20% edge addition.

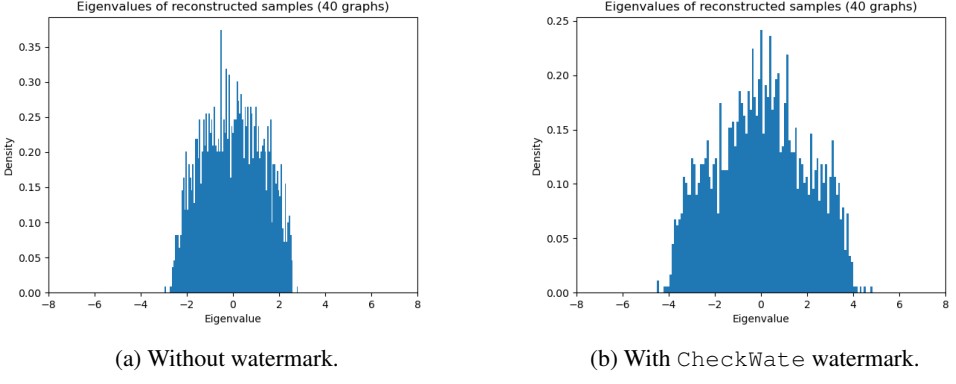

(a) Without watermark.

(b) With `CheckWate` watermark.

Figure 18: Distribution of reconstructed eigenvalues on Tree dataset under 20% node deletion.

