# OpenReview forum: "CheckMate! Watermarking Graph Diffusion Models in Polynomial Time"
_ICLR.cc/2026/Conference — ICLR 2026 Poster_

### Official Review · Reviewer_KeKm · 2025-10-14

**Soundness:** 3
**Presentation:** 3
**Contribution:** 3
**Rating:** 6
**Confidence:** 3

**Summary:**

This paper proposes CheckWate, a novel sampling-time watermarking framework for graph diffusion models. By embedding checkerboard patterns into the eigenvalues of noisy latent representations, CheckWate achieves isomorphism-invariant, polynomial-time watermark verification. To enable detection from quantized (i.e., discrete) graphs, the authors develop an approximate dequantization method based on spectral properties, and introduce a latent sparsification mechanism to suppress false positives under adversarial graph perturbations. The method is thoroughly evaluated across multiple synthetic and real-world graph datasets under diverse attacks, showing strong generation quality and watermark detectability.

**Strengths:**

- This is the first paper to explore watermarking in graph diffusion models, extending prior image-based techniques to a much more complex domain.
- The use of graph eigenvalues as watermark carriers is well-motivated by their isomorphism invariance, and the checkerboard embedding leverages known spectral laws to ensure detectability.
- The approximate dequantization and sparsification modules are theoretically grounded and practically effective, enabling detection even under isomorphism and heavy graph edits.

**Weaknesses:**

- Z-score does not reflect full detection performance
While Z-score is useful for summarizing distributional separation, it does not directly reflect practical detection metrics such as true positive rate (TPR), false positive rate (FPR), or ROC curves. As a reviewer, I would prefer to see:
    - Detection threshold selected to ensure low FPR (e.g., 1%), then report TPR.
    - Alternatively, a ROC/AUC analysis on the per-graph detection scores.
This is especially relevant for real-world deployment or fair method comparison.

- The use of “2” as the bulk-blip threshold lacks justification
The watermark detection pipeline considers eigenvalues with magnitude >2 to be outside the bulk (Line 237). While this is motivated by the Wigner semicircle law, the choice of 2 as a hard threshold can be brittle since real data distributions are finite and noisy.
I recommend reporting sensitivity analysis of Z-scores under varying thresholds (e.g., 1.8/2.2) or using quantile-based definitions.

- Tradeoff between watermark strength and dequantization accuracy is underexplored
According to Line 295 and Theorem 3.3, the dequantization error increases with eigenvalue multiplicity, which itself depends on watermark strength (larger W, smaller k). This raises a natural question: Does stronger watermarking hurt detectability due to worse latent reconstruction? The authors should analyze this tradeoff explicitly, possibly via a plot of detectability vs. reconstruction error for varying k and W.

- Line 360 has a type of "Appendix ??".

**Questions:**

Please refer to the weakness part. In short,
- Can you report ROC/AUC curves or TPR/FPR under fixed thresholds, in addition to Z-scores?
- Why was 2 chosen as the cutoff between bulk and blip eigenvalues? Have you tested threshold robustness?
- How does watermark strength (via k and W) affect the tradeoff between detectability and dequantization error?

---

> ### Author Response · Authors · 2025-11-17
> **Answer to comments.**
>
> We thank the reviewer for the positive feedback to our work. We appreciate the reviewer's acknowledgment of our effort in building a theoretically grounded framework.
>
> We hereby address Reviewer KeKm's comments. The requested ablation studies enabled us to further strenghten the theoritcal groundness of CheckWate.
>
> ### [QW1] **ROC/AUC curves for True and False Positive Rates**
>
> [AQW1] We reran the experiments of Table 1 and 2. In the revised manuscript we provide the requested ROC curves for the Planar and SBM datasets (the latter one being the dataset with the lowest z-scores) with four different attacks at their maximum strength:
>
> **Planar** Figure 4 (Appendix G)
>
> **SBM** Figure 5 (Appendix G)
>
> The figures show that CheckWate consistently achieves a AUC of 1.0 under all combinations. Meaning CheckWate always manages to achieve a TPR of 100% and a FPR of 0%.
>
> In answer [WQ2] to reviewer kR4u, we further provide the ROC figures without the latent sparsification method. Showcasing that the latter component is key for providing robust detection under major perturbations.
>
> ### [QW2] **Use of 2 as bulk threshold**
>
> [AQW2]Indeed, there is a possibility of noise around 2. We mention that eigenvalues should be $\gg 2$ at line 237 of the paper. We will further underline this aspect to make it clearer.
>
>
> If interested, in Appendix F we provide a qualitative analysis of the eigenvalues distributions in watermarked and non-watermarked data under different types of perturbations.
> While eigenvalues of non-watermarked graphs can be larger than 2, the spectrum of watermarked latents have values of larger magnitude.
>
> Further, we show numerically that the noise is not enough to impact CheckWate detectability. We measure the average of the largest $k$ squared eigenvalues under Planar for 40 watermarked and non-watermarked graphs. Then, we report minimum and maximum results. The result show that the largest eigenvalues of non-watermarked remain close enough to 2, and that CheckWate eigenvalues are much larger and easily separable.
>
> **Planar**
> |Watermark | min | max|
> |-|-|-|
> |None | 3.64 | 4.35 |
> |CheckWate | 9.7 | 10.995|
>
>
> ### [QW3] **Experiments with different $k, W$**
> [AW3] Here, we widen our results with an extensive analysis of how $k$ and $W$ can affect Z-score and generative quality. For such results, please see the tables in Appendix F of the revised paper. The key observations are, as expected, generation quality degrades for larger values of $W$ and smaller $k$. Accordingly, Z-score increase when $W$ is smaller and $k$ larger. This confirms the theory discussed in the main content of the paper.
>
> **Planar** Table 5 in the revised paper (Appendix F)
>
> **Tree** Table 6 in the revised paper (Appendix F)
>
> **SBM**  Table 7 in the revised paper (Appendix F)
>
> ### [W4] **Typos and proofreading**
>
> [AW4] We thank the reviewer for noting this. We will thoroughly proofread the paper in our next revision.

---

> ### Author Response · Authors · 2025-11-28
>
> Dear Reviewer KeKm,
>
> If we resolved your concerns, please consider raising our score. If you still have questions or concerns, please let us know.
>
> Kind regards,
>
> The Authors

---

### Official Review · Reviewer_8DVk · 2025-10-22

**Soundness:** 3
**Presentation:** 3
**Contribution:** 3
**Rating:** 6
**Confidence:** 2

**Summary:**

This paper introduces CheckWate, a watermarking framework for graph diffusion models that enables polynomial-time watermark verification. The method embeds a checkerboard pattern in the eigenvalues of the latent diffusion space.  The benefit of this approach is that eigenvalues are isomorphism-invariant. Prior techinques based on graph isomorphism (or graph edit distance) run into the issue that these problems are NP-hard. This approach avoids that issue.

Experiments on four datasets show strong performance under various attacks (isomorphism, edge/node perturbations), outperforming prior diffusion watermarking methods such as Gaussian Shading and TreeRing.

**Strengths:**

This is a watermarking method specifically designed for graph diffusion models, a growing area in generative AI and data governance.

There are mathematical results with proofs.

There's a good set of experiments: multiple datasets, attacks, comparison approaches.

**Weaknesses:**

Assumptions (continuous latent spaces?).  (The paper states it could be applied to discrete latent spaces in Appendix D, but doesn't discuss fully.)

Watermark is non-blind.

Polynomial time here is O(N^3), which is non-trivial;  accordingly, it seems, experimental tests are for graphs with less than 500 nodes. More information on running time would be useful.

Some writing things to check (I see ?? for missing references, e.g., Appendix ??, please check).

**Questions:**

What is the empirical running time for watermark detection?

---

> ### Author Response · Authors · 2025-11-17
> **Answer to comments.**
>
> We thank the reviewer for the positive feedback. We appreciate the reviewer acknowledging the impact, the theoretical strength, and the extensive evaluation of our method.
>
> We hereby address Reviewer 8DVk comments. 8DVk's comments enabled us to start discussions on interesting possible future research directions. These comments include an exciting network science-related discussion on the spectrum of random discrete matrices.
>
> ### [W1] **How about discrete latent space**
>
> [AW1] Indeed, as correctly identified by the reviewer, CheckWate requires to use a continuous latent representation. The key idea of CheckWate watermark lies in moving some of the eigenvalues outside of the bulk (i.e., Wigner semicircle) while applying minimal changes to the latent. In the context of continuous latents, we achieve this by applying the checkered entries.
>
> **Discrete noisy latents** We believe that the same idea can be applied in discrete space in the following way.
> **Eigenvalues of discrete noisy latents** (i.e., Erdos-Renyi matrices=independent binomial) also follow the Wigner semicircle law. Previous studies [1] showed that the presence of eigenvalues in the blip (i.e., outside of the semicircle) is related with the presence of communities. We suggest that enforcing the presence of communities in parts of the noisy latent can be used to reproduce CheckWate behavior in the discrete scenario. The main bottleneck to overcome in discrete latent space is reverting the diffusion process to reconstruct the original latent. While we manage to reconstruct continuous latents thanks to our dequantization mechanism, we leave the discrete counterpart for future work.
>
> ### [W2] **Non-blindness**
>
> [AW2] Indeed, as correctly identified by the reviewer, we do not provide a blind watermarking technique. We would like to emphasize that none of the available graph watermarking techniques in the literature are blind. We find addressing computational feasibility of non-blind graph watermarking is a key step to enable future development in this field.
> We recognize this is an interesting direction for future work. We will stress more this in the paper content and the limitations section.
>
> ### [W3+Q1] **Time complexity**
>
> [AW3Q1] In our experiments we measured the following time consumption:
>
> | Dataset (Nodes)| Verification (per graph)| Reverse Diffusion (40 graphs, 200 steps)|
> |-|-|-|
> | Planar (64 nodes)    | ~0.0068 s | ~14.8 s |
> | SBM (192 nodes)      | ~0.0078 s                 | ~60 s                                     |
> | Proteins (500 nodes) | ~0.0081 s                 | ~200 s                                    |
>
> While eigendecomposition takes $\mathcal O(N^3)$, we argue that the performance bottleneck is training of the diffusion model. All of our models are pretrained from publicly available code of GruM [3] except for the Tree dataset (64 nodes). Training the Tree dataset took us roughly 8 hours on one Nvidia L40S.
>
> Scalability of graph diffusion model is currently a topic of major interest in the research community [2,3].
>
> ### [W4] **Writing and proofreading**
>
> [AW4] We thank the reviewer for noting this. We will thoroughly proofread the paper in our next revision.
>
> ### **References**
>
> [1] Budel, Gabriel, and Piet Van Mieghem. "Detecting the number of clusters in a network." Journal of Complex Networks 8.6, 2020.
>
> [2] Luo, T., et al. "Fast graph generation via spectral diffusion." IEEE Transactions on Pattern Analysis and Machine Intelligence 46.5 (2023).
>
> [3] Fu, X., et al. "Hyperbolic geometric latent diffusion model for graph generation." ICML, 2024.

---

> > ### Comment · Reviewer_8DVk · 2025-11-25
> >
> > I have seen the responses given by the authors to my and other reviews.  I appreciate their work.  I am planning to keep my score the same.

---

> > > ### Author Response · Authors · 2025-11-26
> > > **Thanks for the reply**
> > >
> > > Dear Reviewer 8DVk,
> > >
> > > Thank you for your response. If any concern will come up, please do not hesitate in asking further questions.
> > >
> > > Regards,
> > >
> > > The Authors

---

### Official Review · Reviewer_kbtq · 2025-10-26

**Soundness:** 3
**Presentation:** 2
**Contribution:** 3
**Rating:** 6
**Confidence:** 2

**Summary:**

This paper proposes CheckWate, the first watermarking framework for graph diffusion models that enables verification in polynomial time. Traditional methods are challenged by the computationally hard Graph Isomorphism (GI) problem. To bypass this, CheckWate embeds a "checkerboard" watermark into the latent eigenvalues, which are isomorphism-invariant. The framework uses an approximate dequantization mechanism to revert the discrete graph to its continuous latent for verification and a latent sparsification method to improve robustness against attacks. Experiments demonstrate that CheckWate maintains high generation quality and is robust against attacks like isomorphism, where baselines fail completely.

**Strengths:**

1.	It introduces CheckWate, the first watermarking framework specifically designed for graph diffusion models that provides exact and polynomial time verification.
2.	The framework demonstrates strong robustness against various graph modification attacks, including isomorphism, edge deletion, edge addition, and node deletion.

**Weaknesses:**

1.	The paper's primary claim of efficiency is supported by theoretical complexity analysis but lacks empirical validation. A direct comparison of wall-clock runtime against baseline methods, particularly on large-scale graphs, would be necessary to fully substantiate the practical performance benefits of the proposed polynomial-time approach.
2.	The paper could benefit from a careful proofreading to correct minor typographical errors (e.g., around line 360 in the provided draft), which would improve the overall clarity and presentation quality.
3.	The non-blind nature of the watermark raises significant scalability concerns. The paper does not address the retrieval problem when the database of original watermarked graphs (c) is large. Verifying a suspect graph would seem to require a linear scan through all c candidates, leading to a total complexity of $c \cdot O(N^3)$, which may be impractical for large-scale auditing or copyright enforcement scenarios.
4.	The paper lacks a sensitivity analysis for the key hyperparameters (k, W). A thorough ablation study is needed to demonstrate how different choices of k and W affect the fundamental trade-off between watermark robustness and generative quality, which is crucial for understanding the method's practical applicability across different datasets and requirements.
5.	The reliance on Z-score as the primary metric for detectability is insufficient and potentially misleading. The paper's own results in Table 2 show the Bipartite baseline achieving Z-scores that are an order of magnitude higher than CheckWate, making the proposed method appear far weaker. To provide a true measure of effectiveness, the authors must report standard classification metrics. Reporting the watermark detection ACC and AUC is essential to clarify whether CheckWate's Z-score, while lower, is already sufficient for near-perfect detection, thus framing Bipartite's superior score as a result of severe, quality-damaging overkill.

**Questions:**

See the weakness.

---

> ### Author Response · Authors · 2025-11-17
> **Answer to weaknesses.**
>
> We thank the reviewer for its positive feedback. We appreciate the reviewer acknowledging how CheckWate improves over the prior art and its robustness to strong attacks.
>
> We hereby address Reviewer kbtq's comments. Reviewer's suggestions enabled us to provide further experimental evidence that supports our theoretical claims.
>
> ### [W1] **Time cost analysis of CheckWate**
> [AW1] In our experiments we measured the following time consumption:
>
> | Dataset (Nodes)| Verification (per graph)| Reverse Diffusion (40 graphs, 200 steps)|
> |-|-|-|
> | Planar (64 nodes)    | ~0.0068 s | ~14.8 s |
> | SBM (192 nodes)      | ~0.0078 s                 | ~60 s                                     |
> | Proteins (500 nodes) | ~0.0081 s                 | ~200 s                                    |
>
>
> While eigendecomposition takes $\mathcal O(N^3)$, we argue that the performance bottleneck is training of the diffusion model. All of our models are pretrained from publicly available code of GruM [3] except for the Tree dataset (64 nodes). Training the Tree dataset took us roughly 8 hours on one Nvidia L40S.
>
> Scalability of graph diffusion model is currently a topic of major interest in the research community [1,2].
>
> ### [W2] **Proofreading**
>
> [AW2] We thank the reviewer for noting this. We will thoroughly proofread and correct the paper in our next revision.
>
> ### [W3] **Non-blindness and computianal costs concerns**
>
> [AW3] Indeed, as correctly identified by the reviewer, we do not provide a blind watermarking technique. Unfortunately, the prior art on graph watermarking techniques assumes non-blindness. We find addressing computational feasability of non-blind graph watermarking is a key step to enable future development in this field.
> We recognize this is an interesting direction for future work. We will stress this more in the paper content and the limitations section.
>
> About the concerns on computational costs, we find that verification of a single graph is generally fast. We discuss this in more details in [AW1].
>
> ### [W4] **Sensitivity analysis for $k, W$**
> [AW4]  Here, we widen our results with an extensive analysis of how $k$ and $W$ can affect Z-score and generative quality. For such results, please see the tables in Appendix F of the revised paper. The key observations are, as expected, generation quality degrades for larger values of $W$ and smaller $k$. Accordingly, Z-score increase when $W$ is smaller and $k$ larger. This confirms the theory discussed in the main content of the paper.
>
> **Planar** Table 5 in the revised paper (Appendix F)
>
> **Tree** Table 6 in the revised paper (Appendix F)
>
> **SBM**  Table 7 in the revised paper (Appendix F)
>
> ### [W5] **ROC curves for True Positive and False Positive analysis**
>
> [AW5] We reran the experiments of Table 1 and 2. In the revised manuscript we provide the requested ROC curves for the Planar and SBM datasets (the latter one being the dataset with the lowest z-scores) with four different attacks at their maximum strength:
>
> **Planar** Figure 4 (Appendix G)
>
> **SBM** Figure 5 (Appendix G)
>
> The figures show that CheckWate consistently achieves a AUC of 1.0 under all combinations. Meaning CheckWate always manages to achieve a TPR of 100% and a FPR of 0%.
>
> In answer [WQ2] to reviewer kR4u, we further provide the ROC figures without the latent sparsification method. This demonstrates that the latter component is key for providing robust detection under major perturbations.
>
> ### **References**
>
> [1] Luo, T., et al. "Fast graph generation via spectral diffusion." IEEE Transactions on Pattern Analysis and Machine Intelligence 46.5 (2023).
>
> [2] Fu, X., et al. "Hyperbolic geometric latent diffusion model for graph generation." ICML, 2024.

---

> ### Author Response · Authors · 2025-11-28
>
> Dear Reviewer kbtq,
>
> If we resolved your concerns, please consider raising our score. If you still have questions or concerns, please let us know.
>
> Kind regards,
>
> The Authors

---

### Official Review · Reviewer_DEc1 · 2025-10-31

**Soundness:** 4
**Presentation:** 4
**Contribution:** 3
**Rating:** 8
**Confidence:** 3

**Summary:**

The authors introduce CheckWate, a novel graph watermarking method for diffusion models. Instead of trying to watermark graph features directly, it is much more efficient to plant the watermark into graph properties that are invariant to isomorphism, i.e., into eigenvalues. This circumvents the watermark detector's need to handle different representations of the same graph, leading to compute performance gains. The authors also introduce dequantization and sparsification mechanisms, which they experimentally validate to be important. The CheckWate system appears to strike a balance between good detection (robustness) and graph quality in comparison to major watermarking systems.

**Strengths:**

The watermarking method is novel and its empirical performance is backed by theoretical underpinnings. Clarity on limitations is to be commended - potential future research paths are important and interesting problems. Paper is well-written and scientific with only a few points of concern (mentioned below).

**Weaknesses:**

- Polytime detector is presented as a major feature of CheckWate, but it is not emphasized in the body. The method appears clearly better than the prior work on graph watermarking that do not circumvent superpolynomial subroutines, but it is unclear how performance compares to the baseline (image) watermarking schemes evaluated.
- I believe graph isomorphsim is known to be in NP (e.g., Babai's quasipolynomial algorithm [1]), but not known to be NP-complete. There are statements in the paper that contradict this, e.g., row 50.
- The selected baselines are predominantly image domain watermarks. There is limited comparison to graph domain watermarks.
- "exact" detection mentioned in the abstract is perhaps misleading. Might be worth rephrasing.

[1] Babai, László. "Graph isomorphism in quasipolynomial time." Proceedings of the forty-eighth annual ACM symposium on Theory of Computing. 2016.

**Questions:**

- CheckMate (instead of CheckWate) in the title - is this intentional?
- How is it possible that Gaussian Shading is provably lossless, yet generation quality is at times worse than lossy watermarks (Table 1)? On row 396: "CheckWate achieves state-of-the-art generative quality" despite it being lossy.
- One of the major upsides of CheckWate seems to be efficient detection (i.e., with respect to "conventional" exponential time approaches). How does the scheme compare concretely to the other baselines in raw compute cost? Apologies if I missed this in the paper.
- Do you expect it to be possible to force keys (K) to be the same? This would lead to forgery attacks. It would be nice to have some kind of threat model to make it clear how "secure" the watermark needs to be. I suspect this is not as much of a concern as it would be for generic mode watermarks since the graph domain may be less adversarial, but I'd like clarity on this.

Presentation remarks
- Row 21/22: approximately dequantizes - > approximately dequantize
- Row 360: Appendix ??

---

> ### Author Response · Authors · 2025-11-17
> **Answer to comments. Part 1**
>
> We thank the reviewer for the positive feedback on our submission. We appreciate the reviewer acknowledging its theoretical groundness and our efforts in clarifying the future directions.
>
> We hereby address the comments by Reviewer DEc1, which we consider insghtful starting points of discussion on the methodology, theoretical background of graph watermarking, and future directions.
>
> ### [Q1] **CheckMate instead of CheckWate in the title**
>
> [AQ1] It is not a mistake. CheckWate stands for **Check**erboard **Wate**rmark, but it is also supposed to be a pun for checkmate (as chess is played on a checkerboard).
>
> M in CheckMate has been capitalized as it is graphically an upside down W.
>
> I am genuinely curious whether the pun is clear to a first-time reader.
>
> ### [Q2] **Gaussian Shading is lossless but CheckWate sometimes performs better**
>
> [AQ2] While Gaussian Shading is a lossless watermarking, this does not necessarily mean that it leads to the best possible generative quality for the following reasons.
>
> Following other works on diffusion watermarking in the literature [1, 2], we use an *implicit* diffusion model (DBIM). Meaning, the diffusion process does not have any noise and is deterministic. On the one hand, this enables the possibility of reconstructing the original latent. On the other hand, the generation quality is reduced due to a lower variance in the generative process.
>
> We hypothesize that applying the checkered entries to the noisy latent increases variance of the noisy latent and partly compensates for the abovementioned loss on some experiments. We further stress that this behavior is not consistent and CheckWate brings generative performance that are overall comparable to the ones of Gaussian Shading.
>
> ### [Q3+W1] **Time complexity**
>
> [AQ3W1] In our experiments we measured the following time consumption:
>
> | Dataset (Nodes)| Verification (per graph)| Reverse Diffusion (40 graphs, 200 steps)|
> |-|-|-|
> | Planar (64 nodes)    | ~0.0068 s | ~14.8 s |
> | SBM (192 nodes)      | ~0.0078 s                 | ~60 s                                     |
> | Proteins (500 nodes) | ~0.0081 s                 | ~200 s                                    |
>
>
> While eigendecomposition takes $\mathcal O(N^3)$, we argue that the performance bottleneck is training of the diffusion model. All of our models are pretrained from publicly available code of GruM [3] except for the Tree dataset (64 nodes). Training the Tree dataset took us roughly 8 hours on one Nvidia L40S.
>
> Scalability of graph diffusion models is currently a topic of major interest in the research community [4,5].
>
>
> ### [Q4] **CheckWate focuses more on computational burden rather than threat model.**
>
> [AQ4] Indeed, as correctly identified by the reviewer, our contributions focuses mostly on making verification on graphs computationally feasible. Watermarking on graphs is a field much less developed than other data modalities (such as images, tables, or time series). Hence, introducing a computationally feasible method is a significant first step for this field, though under a weaker threat model.
>
> ### [Q4.1] **Is it possible to force multiple graphs to have the same $K$?**
>
> [AQ4.1] Forcing multiple $K$ might require the adversary to manipulate the diffusion process such that the non-discretized graph has a certain distribution of eigenvectors and eigenvalues.
>
> If the model is owned by a party different from the malicious data owner (as in most watermarking papers [6]), we would exclude this is possible at all, as the adversary has no control at all over the diffusion process.
>
> Even in the case in which the adversary has access to the model, we argue that manipulating eigenvectors and eigenvalues of the final generated graph is a non-trivial task. Let alone doing it without significantly degrading the data quality.

---

> ### Author Response · Authors · 2025-11-17
> **Answer to comments. Part 2**
>
> ### [W2] **Graph isomorphism in NP**
>
> [AW2] Indeed, graph isomorphism is in NP as a solution can be verified in polynomial time. We meant that it is not yet known whether it can be solved in polynomial time.
>
> We will rephrase the sentence as: "as it is not known whether it can be solved in P." and add the suggested reference. Further, we would like to stress that in the context of graph watermarking, computational hardness is related with Graph Edit Distance, which is NP-hard.
>
> ### [W3] **Comparison with graph watermarks**
>
> [AW3] We considered comparing CheckWate with other graph watermark techniques. However, we choose Gaussian Shading and TreeRing as baselines due to the following reasons:
>
>    * **Post-editing vs Latent** We find more natural to compare in-latent watermark with other in-latent watermarks rather than post-editing method. This aligns with latent-watermarking studies from the literature including [1,2]. Moreover, post-editing tends to degrade data quality more and being more susceptible to adversarial attacks.
>
>    * **Assumptions on data** Graph watermarking baselines make assumptions on the data which prevents them from being applied on some experiments. For instance, [7] requires sufficient node degree heterogeneity. This means it cannot be applied on SBM, where the expected degree is the same for every node of the graph.
>
>    * **Gaussian Shading is a strong baseline**. (If we ignore isomorphism and dequantization) Gaussian Shading is a successful (if not outstanding) watermarking technique for graphs. Looking at Table 1, it brings lossless generative quality and strong detectability (which remains even under attack when using ideal dequantization, Table 3). Altogether, if isomorphism could not be applied on graphs, Gaussian Shading would be a complete and elegant state-of-the-art methodology. In Table 1, matching the performance of Gaussian Shading showcases that CheckWate achieves the highest standards we can currently ask from a watermarking method.
>
> ### [W4] **On "exact" extraction**
>
> [AW4] We recognize that "exact" could be misinterpreted. We used this term to emphasize that we do not rely on greedy approaches to overcome NP-hardness but rather circumvent it using inherent properties of the latent matrix.
>
> We will rephrase the sentence as: "providing polynomial time verification".
>
> ### **References**
>
> [1] Zhu, C., et al. "Tabwak: A watermark for tabular diffusion models." ICLR, 2025.
>
> [2] Soi, ZW, et al. "TimeWak: Temporal Chained-Hashing Watermark for Time Series Data." NeurIPS, 2025.
>
> [3] Jo, J., wt al. "Graph generation with diffusion mixture." ICML, 2024
>
> [4] Luo, T., et al. "Fast graph generation via spectral diffusion." IEEE Transactions on Pattern Analysis and Machine Intelligence 46.5 (2023).
>
> [5] Fu, X., et al. "Hyperbolic geometric latent diffusion model for graph generation." ICML, 2024.
>
> [6] Yang, Z., et al. "Gaussian shading: Provable performance-lossless image watermarking for diffusion models." CVF Conference on Computer Vision and Pattern Recognition. 2024.
>
> [7] Zhao, X., et al. "Towards graph watermarks." Proceedings of the 2015 ACM on Conference on Online Social Networks. 2015.

---

> ### Author Response · Authors · 2025-11-28
>
> Dear Reviewer DEc1,
>
> If we resolved your concerns, please consider raising our score. If you still have questions or concerns, please let us know.
>
> Kind regards,
>
> The Authors

---

### Official Review · Reviewer_kR4u · 2025-11-01

**Soundness:** 3
**Presentation:** 3
**Contribution:** 3
**Rating:** 6
**Confidence:** 4

**Summary:**

The paper proposes a new graph watermarking approach CheckWate with three main contributions:
(1) a checkerboard watermark technique that embeds signals into noisy latent eigenvalues; since eigenvalues are isomorphism-invariant, detection can be done in polynomial time without solving NP-hard GI/GED problems;
(2) an approximated dequantization mechanism that projects discrete graphs to a continuous space for accurate latent reconstruction and watermark verification;
(3) a robust detection mechanism aimed at reducing false positives.

The experiments show that CheckWate can improve watermark detectability while maintain the graph quanlity.

**Strengths:**

* The paper is well written, logically structured, and easy to follow. The problem is important and well motivated for graph diffusion watermarking.
* Leveraging eigenvalues for watermark embedding/detection is a reasonable way to bypass GI/GED and keep verification polynomial.
* The method addresses graph watermark embedding, approximate dequantization, and robust detection in one framework, largely solving the targeted problem at a conceptual level.

**Weaknesses:**

* Sec. 3.1 states that for high enough $k$, checkerboard ensembles approximate regular Gaussian noise "while forcing limited modifications", but there is no quantitative analysis of how does the distribution change with $k$. Since regular Gaussian noise is a key assumption for diffusion latents, this gap weakens the claim.
* Sec. 3.4 argues that eigenvalues of perturbed GOE may leave the bulk and cause false positives and proposes sparsification to fix this, but experiments do not present the baseline FPR (without robust detection) versus post-fix FPR to show the reduction magnitude.
* An important baseline, Bipartite, is used but lacks a clear literature reference and detailed explanation in the main text; given its very high Z-scores yet relatively low quality metrics, it would be informative to see tuned variants that trade detectability for better quality, potentially making it a stronger baseline.
* The paper states that disabling the robust detection mechanism reduces Z-scores, but comparing Table 2 and Table 4 shows both increased and decreased Z-scores, which is inconsistent with the paper's statement.
* Line 360: “Appendix ??” appears in the draft and should be fixed.

**Questions:**

1. Please report quantitative divergence between checkerboard ensembles and Gaussian across $k$. Where is the operational $k$ range in which deviations remain “limited”?
2. What false positive rates arise from perturbed GOE without the robust step, and what are the FPRs with it? A per-dataset breakdown would substantiate the robustness claim.
3. Please add a citation and methodological details for the baseline Bipartite, and provide quality-controlled tuning where detectability is traded for higher quality to assess how strong Bipartite can be under matched quality.

---

> ### Author Response · Authors · 2025-11-17
> **Answers to comments. Part 1**
>
> We thank the reviewer for the positive feedback. We appreciate the reviewer acknowledging the writing quality, the novelty, and the impact of our manuscript. We further appreciate the reviewer noting that we solve the problem in a reasonable and conceptual level.
>
> We hereby address the comments by Reviewer kR4u. The requested ablation studies and clarifications will further strengthen the results included in the main paper.
>
> ### [WQ1] **How many checkered entries in the noisy latent?**
>
> [AWQ1] In our original experiments, we used the following parameters:
>
> **Planar**: k=90%*N=58, W=$\pm 2.5$
>
> **Tree**: k=80%*N=52, W=$\pm 2$
>
> **SBM**: k=70%*N=134, W=$\pm 2.5$
>
> **Proteins**: k=20%*N=100, W=$\pm 2.5$
>
> which means 0.3%, 0.6%, 0.6%, and 0.8% of the latent entries are checkered.
> This means we edit a low number of entries in the latent, which enables us to maintain high generative quality.
>
> ### [WQ1.1] **How does $k$ change generation quality?**
>
> [AWQ1.1] Here, we widen our results with an extensive analysis of how $k$ and $W$ can affect Z-score and generative quality. For such results, please see the tables in Appendix F of the revised paper. The key observations are, as expected, generation quality degrades for larger values of $W$ and smaller $k$. Accordingly, Z-score increase when $W$ is smaller and $k$ larger. This confirms the theory discussed in the main content of the paper.
>
> **Planar** Table 5 in the revised paper (Appendix F)
>
> **Tree** Table 6 in the revised paper (Appendix F)
>
> **SBM**  Table 7 in the revised paper (Appendix F)
>
>
> ### [WQ2] **Robust detection mechanism and false positive rate**
>
> [AWQ2] We reran the experiments of Table 1 and 2. In the revised manuscript we provide the requested ROC curves for the Planar and SBM datasets (the latter one being the dataset with the lowest z-scores) with four different attacks at their maximum strength:
>
> **Planar** Figure 4 (Appendix G)
>
> **SBM** Figure 5 (Appendix G)
>
> All figures show an AUC of 1.0, which means CheckWate always manages to achieve 100% TPR and 0% FPR.
>
> We further provide the same ROC curves **without the latent sparsification mechanism**:
>
> **Planar** Figure 6 (Appendix G)
>
> **SBM** Figure 7 (Appendix G)
>
> From the plots, we can see that under strong edge additions, the AUC of CheckWate reduces severely: 0.5 for Planar and 0.75 for SBM. A result that indicates little to no discriminatory capability.
>
> Furthermore, we would like to highlight the figures in Appendix H of the paper, in which we perform qualitative analysis on the eigenvalue distributions of our experiments.
> Especially, Fig. 8 showcases how the eigenvalues of the reconstructed latents tend to explode under edge additions (if no latent sparsification is applied).
>
> Altogether, these results show that: (i) CheckWate provides remarkable performance on TPR and FPR; (ii) Latent sparsification is key for preventing performance degradation under major perturbations.

---

> ### Author Response · Authors · 2025-11-17
> **Answers to comments. Part 2**
>
> ### [WQ3] **Details on Bipartite baseline**
>
> [AWQ3] Bipartite is a baseline we developed. We will clarify this in the main paper and provide more details in Appendix B.
>
> In Bipartite, the starting latent is generated via distribution preserving sampling from the adjacency matrix of a complete bipartite graph. When the latent is reconstructed, we discretize it as
> $$
> L =\\begin{cases}0 & \\text{if  } \\Phi(L) < 0.5 \\\\ 1 & \\text {otherwise} \\end{cases}
> $$
> and compute spectral bipartivity [1].
> To increase its robustness, we leverage monotonicity to sample a subgraph of L
> $$
> L =\\begin{cases}0 & \\text{if  } \\Phi(L) < 0.75 \\\\ 1 & \\text {otherwise} \\end{cases}
> $$
> Recall that monotonicity means that a subgraph of a bipartite graph is always bipartite.
>
> The poor performance of Bipartite can be explained by the presence of correlation between entries of the noisy latent. This correlation is inherently caused by the properties of bipartivity which splits nodes into two subsets and connects them accordingly.
>
> ### [W4] **Some Z-scores are lower after latent sparsification**
>
> [AW4] Applying the robust detection mechanism leads to modifications in the latent. As sparsification (roughly) leads to smaller eigenvalues it can happen that the statistical signifcance of CheckWate decreases.
>
> However, we find that the gains obtained by robust detection severely outweight the losses. Taking the results from [AWQ2], the "weaker" Z-scores are more than enough to achieve an AUC of 1.0. In the meantime, by applying this mechanism we lead to significant improvement in resistance to perturbations such as edge additions.
>
> ### [W5] **Proofreading**
>
> [AW5] We thank the reviewer for noting this. We will thoroughly proofread and correct the paper in our next revision.
>
> ### **References**
>
> [1] Estrada, E., and Rodríguez-Velázquez, J. A.. "Spectral measures of bipartivity in complex networks." Physical Review E, 2005.

---

> > ### Comment · Reviewer_kR4u · 2025-11-27
> >
> > Thank you for the comprehensive rebuttal.
> >
> > The authors have responded to my questions and clarified some technical details. I did not see new concerns raised by the response. I'll keep my current positive score.

---

> > > ### Author Response · Authors · 2025-11-27
> > > **Thank you for your response**
> > >
> > > Dear Reviewer kR4u,
> > >
> > > Thank you for your response and positive feedback. If any concern will come up, please do not hesitate in asking further questions.
> > >
> > > Regards,
> > >
> > > The Authors

---

### Author Response · Authors · 2025-11-17
**Rebuttal summary.**

We thank all reviewers for the valuable and positive feedback. We genuinely appreciate the general consensus in finding CheckWate theoretically grounded, novel, and impactful.

We conducted additional experiments and added new figures to address the comments. All the updates will be added to the revised paper.

**Summary of additional experiments:**
* Ablation studies on different values for $(k, W)$. (Rev: kR4u, kbtq, KeKm)
* ROC plots to show True Positive and False Positive ratio. (Rev: kR4u, kbtq, KeKm)
* ROC plots without the latent sparsification mechanism. (Rev: kR4u)
* Measures computational time for watermark verification. (Rev: kbtq, 8DVk)

Furthermore, we addressed clarifications, limitations, and future work:
* Deviation from standard Gaussian noise (Rev: kR4u)
* Details on Bipartite baseline (Rev. kR4u)
* Effect of latent sparsification on detectability (Rev: kR4u)
* CheckWate (sometimes) outperforming Gaussian Shading in generation quality (Rev: DEc1)
* Threat model (Rev: DEc1)
* Theoretical hardness of isomorphism (Rev: DEc1)
* Comparison with prior art in graph watermarking (Rev: DEc1)
* Non-blindness of CheckWate (Rev: kbtq, 8DVk)
* Discrete latent space (Rev: 8DVk)
* Use of 2 as the bulk threshold (Rev: KeKm)

---

### Comment · Area_Chair_yJkU · 2025-11-20
**Form of the Rebuttal**

Dear Authors,

Thank you for your responses. The submitted paper should be updated instead of giving links to external pages in your responses. Please, update your submitted work and give references to Tables and Figures in your updated paper.

With kind regards,

Your AC

---

> ### Author Response · Authors · 2025-11-21
> **Revised Manuscript**
>
> Dear AC and Reviewers,
>
> We uploaded the revised manuscript as kindly requested by the AC and replaced external links with references to the figures and tables of paper.
>
> To the revised manuscript we made the following additions:
> * Ablation study for different values of $k, W$ (Tables 5-7, Appendix F)
> * ROC curves of CheckWate (Figures 4-8, Appendix G)
> * Extended limitations section to include more a detailed discussion on discrete noisy latents (Appendix E)
> * Clarified the implementation details of Bipartite (Appendix B) and clarified that it is a baseline designed by us (Sec 4)
> * Removed "exact" from the abstract
> * Clearly mention that CheckWate is non-blind in our contributions (Sec 1) and that state of the art graph watermarking is non-blind (Sec 2)
> * Clarified relationship between isomorphism and NP-hardness and added the suggested reference (Sec 1)
> * Revised the noted typos
>
> With kind regards,
>
> The authors

---

### Meta-Review · Area_Chair_528W · 2026-01-06

**Summary:**

This paper proposes CheckWate, a watermarking method for graph diffusion models, embedding watermarks into eigenvalues (isomorphism-invariant) to achieve polynomial-time verification. It introduces a theoretically justified approximate dequantization and latent sparsification to handle discrete graphs and robustify detection.

he paper targets an important and underexplored problem of watermarking for graph generative diffusion models. The design choice of watermarking spectral quantities (eigenvalues) is principled and directly addresses isomorphism robustness, providing a coherent path to polynomial-time verification. Solid theoretical and empirical analysis demonstrate its effectiveness.

Initially, reviewers (kR4u, kbtq, 8DVk, KeKm, DEc1) raised concerns regarding:
1. Empirical runtime validation of polynomial-time claims
2. Sensitivity analysis of hyperparameters
3. Robust detection metrics beyond Z-scores (ROC/AUC)
4. Clarity on baseline details and complexity discussions

The authors effectively resolved these issues through additional ablations, explicit runtime measurements, ROC/AUC evaluations, and manuscript clarifications, with reviewers indicating satisfaction.

**Reviewer Concerns:**

The initial concerns are not significant and have been well resolved.

**Reviewer Scores:**

Reviewer kR4u: Likely unchanged. The reviewer explicitly stated that all technical concerns were addressed and no new issues were raised, and they chose to keep their positive score.

Reviewer DEc1: Unchanged. This reviewer already provided a strong accept recommendation and did not raise further concerns after the authors’ clarifications.

Reviewer kbtq: Likely unchanged. The main concerns regarding runtime validation, ROC/AUC metrics, and hyperparameter sensitivity were directly addressed with new experiments, though the reviewer did not explicitly confirm a score change.

Reviewer 8DVk: Unchanged. The reviewer acknowledged the responses and indicated they planned to keep their original score.

Reviewer KeKm: Likely unchanged. The reviewer’s key concerns (ROC/AUC, threshold robustness, and parameter tradeoffs) were fully addressed in the revised manuscript, with no follow-up objections.

All reviewers provided a positive initial assessment, placing the paper clearly above the acceptance threshold from the outset. Given that their initial concerns were minor and subsequently addressed, their scores are expected to remain unchanged.

---

### Decision · Program_Chairs · 2026-01-26

Accept (Poster)